


# The critical role of oxygenated volatile organic compounds (OVOCs) in shaping photochemical O₃ chemistry and control strategy in a subtropical coastal environment

Lirong Hui[1], Yi Chen[1,2], Xin Feng[1], Hao Sun[1], Jia Guo[3], Yang Xu[1], Yao Chen[1], Penggang Zheng[1], Dasa Gu[1] and Zhe Wang[1,*]

[1] Division of Environment and Sustainability, The Hong Kong University of Science and Technology, Hong Kong SAR, 999077, China

[2] Department of Chemistry, The Hong Kong University of Science and Technology, Hong Kong SAR, China

[3] Environmental Central Facility, Institute of Environment, The Hong Kong University of Science and Technology, Hong Kong SAR, 999077, China

*Correspondence to*: Zhe Wang: z.wang@ust.hk

## Abstract

Photochemical ozone (O₃) pollution remains a persistent environmental challenge, and growing evidence highlights the critical role of oxygenated volatile organic compounds (OVOCs) in photochemical processes. However, comprehensive and quantitative measurements of OVOCs remain limited. This study investigates the impact of OVOCs on O₃ formation mechanisms and radical budgets by intergrating high-resolution field measurements from a subtropical coastal region in South China with observation-based photochemical modeling. 63 OVOC species were quantified by a proton-transfer-reaction time-of-flight mass spectrometry (PTR-ToF-MS), and account for 72%-77% of total VOC concentrations. The O₃-precursor relationship analysis revealed a transition regime for O₃ formation and high sensitivity to OVOCs. OVOC-related reactions, including OVOC photolysis, OVOC oxidation by OH and NO₃ radicals, contributed approximately 36%-73% to daytime production rates of HO₂ and RO₂ radicals. Model simulations without comprehensive consideration of OVOCs would significantly underestimate daytime production rates of O₃ and ROₓ radicals by 41%-48%, and shift the diagnosis of O₃ formation from a transition regime to a VOC-limited regime, leading to biased policy recommendations and potentially ineffective control strategies. These findings underscore the critical role of OVOCs in atmospheric photochemistry and highlight the urgent need for comprehensive OVOC quantification to improve OVOC-inclusive model frameworks.



Such improvements are essential for accurately characterizing $O_3$-precursor relationships and for developing effective and sustainable strategies to mitigate regional $O_3$ pollution.

## 1. Introduction

Ground-level ozone ($O_3$) is a significant secondary air pollutant and a major component of photochemical smog, posing serious threats to human health, ecosystems, and the climate (Feng et al., 2019; Yue and Unger, 2014; Mills et al., 2018). Elevated $O_3$ levels remain a persistent environmental challenge in many urban regions worldwide, especially a notable upward trend in East Asia (Li et al., 2019b; Li et al., 2020). $O_3$ formation in the troposphere arises from complex photochemical reactions involving nitrogen oxides ($NO_x$) and volatile organic compounds (VOCs) under the sunlight (Zhao et al., 2022; Xu et al., 2022). The oxidation of VOCs by hydroxyl (OH) radical plays a central role in producing peroxy radicals, such as hydroperoxyl ($HO_2$) and alkyl peroxy ($RO_2$) radicals, which sustain the chain reactions driving photochemical $O_3$ production (Lyu et al., 2022).

Previous studies have underscored the importance of non-methane hydrocarbons, particularly alkenes and aromatics, as major precursors of $O_3$ formation, leading to targeted control measures in various regions (Li et al., 2015; Li et al., 2017; Hong et al., 2019). However, recent research highlights the crucial roles of oxygenated VOCs (OVOCs) in regulating atmospheric oxidation capacity and contributing to radical production (Wang et al., 2022a; Shen et al., 2021; Chai et al., 2023; Yang et al., 2023). OVOCs, such as carbonyls and alcohols, can be emitted directly from diverse sources or formed as secondary products from VOC oxidation (Mellouki et al., 2003). These OVOC compounds can further react with OH radical or undergo photolysis processes, serving as significant sources of $HO_2$ and $RO_2$ radicals that amplify radical cycling and promote $O_3$ formation (Xue et al., 2016; Chen et al., 2020; Huang et al., 2020b). Despite their importance, the diversity and high reactivity of OVOCs introduce substantial uncertainties in atmospheric chemistry and air quality models, particularly in regions with limited OVOC measurements.

Traditional analytical techniques such as gas chromatography (GC) coupled with flame ionization or mass spectrometry detection (FID/MS) have been widely used to measure non-methane hydrocarbons, but only a subset of OVOCs (Huang et al., 2015; Yang et al., 2019; Li et al., 2019a; Han et al., 2019). High-performance liquid chromatography (HPLC) can detect several carbonyl compounds, such as formaldehyde, acetaldehyde, acetone, but its reliance on offline sampling limits its temporal resolution, making it less effective for capturing real-time atmospheric variations (Lu et al., 2010; Yang et al., 2017; Zhang et al., 2024). Furthermore,





many other key OVOC species, such as larger aldehydes, ketones, carboxylic acids, organic peroxides, and other multifunctional compounds, have been rarely measured and poorly characterized in ambient air, which may result in underestimating the role of OVOCs in atmospheric chemistry (Wang et al., 2022a). While $NO_x$ and VOCs are well-recognized as the

primary drivers of $O_3$ formation, the role of OVOCs in shaping photochemical $O_3$ chemistry has received comparatively less attention due to limited field observations and insufficient representation in chemical models. In addition, previous model studies tended to considerably underestimate $RO_x$ (OH, $HO_2$ and $RO_2$) radicals compared to observation (Hofzumahaus et al., 2009; Ma et al., 2019; Rohrer et al., 2014). Some studies have attempted to address these gaps

by simulating unmeasured OVOC species using photochemical box model, but large uncertainties still exist, largely due to missing OVOC primary sources, incomplete or underestimated secondary chemical pathways (Karl et al., 2018; Mo et al., 2016; Bloss et al., 2005; Li et al., 2014). These knowledge gaps hinder an accurate understanding of the $O_3$-precursor relationship, complicating the development of effective control strategies.

80        To gain a comprehensive understanding of the role of OVOCs in photochemical $O_3$ chemistry, a continuous field campaign was conducted at a coastal suburban site in Hong Kong in South China. High-resolution measurements of more than sixty OVOC species were measured using proton-transfer-reaction time-of-flight mass spectrometry (PTR-ToF-MS). By integrating these measurements into a model simulation framework, we quantified the

contributions of OVOCs to radical production and $O_3$ formation, and examined their impacts on $O_3$-precursor relationship, providing critical insights into the formulation of targeted strategies for mitigating $O_3$ pollution in this subtropical region and similar environments.

## 2. Methodology

### 2.1 Field measurement and instrumentation

90        Field measurements were conducted at a suburban coastal site (22.33°N, 114.27°E) located at the campus of The Hong Kong University of Science and Technology (HKUST) in eastern Hong Kong. Situated on a cliff overlooking the sea, the site is near a hotel and a construction project, which may introduce influences from construction activities, household activities, and vehicle emissions. The continuous field campaign spanned from September 4 to

December 20 in 2021, covering three seasons: summer (September 4 - October 12), autumn (October 13 - December 1), and early winter (December 2 - December 20). Seasonal divisions was based on the timing of the first synoptic event, as detailed in our previous studies (Feng et



al., 2023). The measurement site is generally affected by long-range regional transport of aged air masses from South and East China due to the Asian monsoon, as well as fresh emission plumes from Hong Kong and the Pearl River Delta region (Ding et al., 2013).


A PTR-ToF-MS (Ionicon Analytik GmbH, Innsbruck, Austria) with $H_3O^+$ as the primary reaction ion was used to measure the gaseous VOC and OVOC species. Ambient air was drawn into the sampling manifold via a 1/16 polyetheretherketone (PEEK) tube at a flow rate of 5 L min$^{-1}$, and a subsample of filtered air (100 mL min$^{-1}$) was directed to the PTR-ToF-

MS inlet, protected by a polytetrafluoroethylene (PTFE) membrane particle filter to exclude dust and debris. The sampling inlet was maintained at 80 °C throughout the measurements to mitigate humidity-related effects, reduce adsorption losses, and ensure gas-phase stability of target compounds prior to entering the drift tube. The instrument operated under optimized conditions: drift tube temperature maintained at 80 °C, drift voltage at 520 V, and drift tube

pressure at 2.8 mbar, achieving a field density ratio (E/N) of 98 Td (1 Td = $10^{-17}$ V cm$^2$). A relatively higher E/N ratio was employed to suppress the formation of water clusters, thereby minimizing the strong humidity dependence of the target species (Yuan et al., 2017).

Automatic mass calibration was conducted every 100 seconds using the built-in Ionicon permeation unit (PerMasCal), which releases strong signals of $m/z$ 203.943 ($C_6H_4I_2H^+$,

fragment) and $m/z$ 330.848 ($C_6H_4I_2H^+$). Background measurement and multi-point calibration were conducted periodically during the field campaign using a Liquid Calibration Unit (LCU, Ionicon) with pure nitrogen and multi-component VOC gas standards. 19 VOC/OVOC standard gases were used for multi-point calibration, achieving linear correlation coefficients ($R^2$) above 0.99 (Table S1). The limit of detection (LOD) for each species was defined as three

times the standard deviation (3σ) of background signal (Zhou et al., 2019), ranged from approximately 0.009 to 0.094 ppbv (Table S1). Transmission correction was applied using a set of reference compounds, including benzene ($m/z$ 79.054), toluene ($m/z$ 93.070), m-xylene ($m/z$ 107.086), 1,2,4-trimethylbenzene ($m/z$ 121.101), dichlorobenzene ($m/z$ 146.976), and trichlorobenzene ($m/z$ 180.937). In total, 117 VOC/OVOC species were identified and

quantified by attributing the measured ion masses to the most plausible molecular contributors, based on established references and prior studies (Yuan et al., 2017; Koss et al., 2018; Wu et al., 2020), as summarized in Table S2. For species lacking calibration standards, concentrations were determined using an assumed proton transfer reaction rate coefficient of $2 \times 10^{-9}$ cm$^3$ s$^{-1}$, combined with mass-dependent transmission correction (Zhang et al., 2022).


However, we acknowledge the several caveats associated with the VOC/OVOC quantification. First, PTR-Tof-MS is limited in its ability to differentiate between isomeric





compounds. Most molecular formulas were therefore evenly distributed among potential isomers (e.g., phenols, nitrophenols), while specific formulas for aldehydes and ketones were identified based on prior studies using GC-PTR-ToF measurement (Koss et al., 2018).

Although we attempted to assign signals based on likely contributors informed by literature, this approach introduces uncertainties in the molecular-level identification. Second, the ions detected by PTR-ToF-MS can include fragmentation products or hydrated clusters, particularly for highly functionalized OVOCs, which may lead to over- or underestimation of specific compounds if not correctly interpreted. These effects are especially relevant for larger or

multifunctional OVOCs that are more prone to fragmentation or clustering. Given these limitations, our quantification of OVOCs should be considered semi-quantitative for uncalibrated species.

In addition, daytime canister samples of VOCs were collected daily in three seasons, and nonmethane hydrocarbons and alkyl nitrates were analyzed using gas chromatograph

system equipped with mass spectrometry, flame ionization, and electron capture detectors (GC-MS/FID/ECD). Trace gases including $O_3$, $NO_x$ (nitric oxide (NO) and nitrogen dioxide ($NO_2$)) and carbon monoxide (CO) were measured by $O_3$ analyzer (Thermo Scientific, model 49i), $NO_x$ analyzer (Ecotech Serinus 40) and CO analyzer (Thermo Scientific, model T300), respectively. The meteorological parameters including temperature (T), relative humidity (RH),

wind speed (WS), and wind direction (WD) were recorded by a Weather Station. Detailed descriptions of the instruments are available in previous work (Hui et al., 2023; Sun et al., 2024).

## 2.2 Photochemistry Modeling

A zero-dimensional photochemical box model (the Framework for 0-D Atmospheric

Modeling, F0AM v4.2.2) coupled with the Master Chemical Mechanism (MCM) v3.3.1 was applied to simulate the atmospheric photochemistry of observed species. The MCM v3.3.1 is a nearly explicit gas-phase chemical mechanism describing over 17000 reactions and 5800 primary, secondary, and radical species. The model simulation was constrained by observed hourly data of meteorological parameters (T, RH, and pressure), trace gases ($O_3$, NO, $NO_2$, and

CO), 38 VOC species measured by GC-MS/FID/ECD (Table S3), and 88 VOC/OVOC species measured by PTR-ToF-MS (Table S2). VOC species from daytime canister samples were linearly interpolated to hourly resolution for the model input (Yang et al., 2018), while nighttime data were approximated using linear regressions of unmeasured concentrations of $C_2$-$C_{10}$ hydrocarbons and alkyl nitrates against continuous measured hydrocarbons (e.g., $C_3H_6$,





$C_5H_{10}$, $C_6H_{10}$) and nitrophenols by PTR-ToF-MS, respectively. These approximations were used primarily to pre-run the model and were not expected to affect daytime simulation results (Chen et al., 2020). Photolysis frequencies within the model were calculated as the function of solar zenith angle (Wolfe et al., 2016). Observation-based simulations were performed for consecutive days with high $O_3$ concentrations during summer (September 8 - October 2),

autumn (November 12 - November 30), and early winter (December 4 - December 19). Three days from summer (September 11, 12, and 17) were selected as the case study of summer high-$O_3$ episode, where the maximum $O_3$ concentration exceeded 110 ppbv. The model was pre-run for four days to stabilize the concentrations of unconstrained species, with results from the $5^{th}$ day used for further analysis.

Model performance was evaluated using the index of agreement (IOA), as illustrated in Equation 1 (Huang et al., 2005), with values of 0.81-0.87 for the simulated $O_3$ across seasons in this study, comparable to previous studies (0.6-0.9), suggesting that the abundance and variation of $O_3$ were deemed reasonably reproduced (He et al., 2019; Liu et al., 2021; Wang et al., 2018b; Wang et al., 2017; Wang et al., 2015).

$$IOA=1-\frac{\sum_{i=1}^{n}(O_i-S_i)^2}{\sum_{i=1}^{n}\left(|O_i-\overline{O}|+|S_i-\overline{O}|\right)^2} \qquad (1)$$

Where $O_i$ and $S_i$ represent the measured and simulated $O_3$ concentration, respectively; $\overline{O}$ represents the mean measured $O_3$ concentration; and n represents the number of samples. The index of IOA typically ranges from 0 to 1, with higher values indicating stronger alignment between simulation and observation.

Dominant photochemical production pathways ($HO_2$ + NO, $RO_2$ + NO) and destruction pathways ($O_3$ photolysis, $O_3$ + OH, $O_3$ + $HO_2$, VOCs + $O_3$, $NO_2$ + OH) of $O_3$ were determined using Equations 2-4 (Tan et al., 2019a; Wang et al., 2018a). The production rates of $RO_x$ radicals (OH, $HO_2$ and $RO_2$ radicals) were also calculated, incorporating primary sources such as photolysis reactions (i.e., $O_3$ photolysis, OVOC photolysis, etc.), VOC reactions with $O_3$

and $NO_3$, as well as recycling processes (Xue et al., 2016; Wang et al., 2018a; Tan et al., 2019b).

To evaluate the model uncertainties associated with the presence of multiple isomers in PTR-ToF-MS measurements, we conducted a sensitivity analysis by estimating the lower and upper limits of $RO_x$ radicals and $O_3$ production rates with possible isomers. In this analysis, each OVOC molecular formula was assigned either to the isomer with the minimum or

maximum photolysis frequencies and $K_{OH}$ values among all plausible isomeric structures. This approach accounts for the variability in chemical reactivity stemming from the unresolved



isomer distribution and provides a range of potential impacts on atmospheric $RO_x$ radicals generation and $O_3$ formation.

$$P(O_3)=k_{HO_2+NO}[HO_2][NO]+\sum k_{RO_2+NO}[RO_2][NO] \tag{2}$$


$$L(O_3)=k_{O(^1D)+H_2O}[O(^1D)][H_2O]+k_{O_3+OH}[O_3][OH]+k_{O_3+HO_2}[O_3][HO_2]$$

$$+\sum k_{O_3+VOCs}[O_3][VOCs]+k_{NO_2+OH}[NO_2][OH] \tag{3}$$

$$\text{Net }P(O_3)=P(O_3)- L(O_3) \tag{4}$$

Where $P(O_3)$, $L(O_3)$, and Net $P(O_3)$ represents the production rate, loss rate, and net production rate of $O_3$, respectively. The $O_3$ photolysis was represented as the reactions of $O(^1D)$

and $H_2O$ (Shen et al., 2021). VOCs here included both constrained VOCs and model simulated VOCs. The constants (k) represent the rate coefficients of each reaction.

The $O_3$-precursors relationship was characterized using the relative incremental reactivity (RIR) method, calculated by Equation 5 (Liu et al., 2021). $O_3$ formation regime was further characterized by $O_3$ isopleth method, which was derived by scaling precursor

concentrations (10%-200% of original values) to simulate $O_3$ concentrations under varying VOCs and $NO_x$ levels (Tan et al., 2018).

$$\text{RIR }(X)= \frac{\left[P_{O_3\text{-}NO}(X)\text{-}P_{O_3\text{-}NO}(X\text{-}\Delta X)\right]/P_{O_3\text{-}NO}(X)}{\dfrac{\Delta S(X)}{S(X)}} \tag{5}$$

Where RIR represents relative incremental reactivity; X represents specific $O_3$ precursor (i.e., VOCs, $NO_x$); S(X) represents observed concentration of precursor X (ppbv); $\Delta S(X)$ represents

hypothetical change of the concentration of precursor X; $P_{O_3\text{-}NO}(X)$ represents net $O_3$ production in a base run with original observed precursor concentrations, while $P_{O_3\text{-}NO}(X\text{-}\Delta X)$ represents the net $O_3$ production in a second run with a hypothetical change $\Delta S(X)$ of 10% in this study. The net $O_3$ production was calculated by Equation 4. A larger positive RIR value indicates higher sensitivity of $O_3$ production to this precursor, implying that reducing emissions

of this precursor would more effectively suppress $O_3$ formation. Conversely, a negative RIR value suggests that emission reductions of this precursor could paradoxically increase $O_3$ production (Wang et al., 2018b).

**3. Results and discussion**

**3.1 Overview of the observations**

During the campaign, 117 VOC/OVOC species were continuously measured using the PTR-ToF-MS, including 2 biogenic VOCs (BVOCs), 24 anthropogenic VOCs (AVOCs,



comprising alkenes, cycloalkanes, and aromatics), 63 OVOCs (categorized as $C_xH_yO_{1-3}$), and 28 nitrogen/sulfur containing VOCs (N/S-containing VOCs). The time series of different VOC groups, meteorological parameters, and trace gases are shown in Figure S1. The campaign

witnessed twenty $O_3$ episode days (maximum $O_3$ value > 80 ppbv) across three seasons, with three extreme episodes exceeding 110 ppbv (defined as high-$O_3$ episodes) in summer. The measured VOCs/OVOCs showed the highest total concentration in early winter (47.84 ppbv), followed by autumn (44.26 ppbv) and summer (28.83 ppbv), which is consistent with the $O_3$ seasonal trend (Figure S4). Furthermore, the total VOCs/OVOCs concentration was much

higher during $O_3$ episode days and reached 60.96 ppbv during summer high-$O_3$ episode days, with 76% contribution from OVOCs, emphasizing their pivotal role in $O_3$ production. OVOCs, AVOCs, and N/S-containing VOCs increased progressively from summer to early winter, with the most pronounced rise observed between summer and autumn (Figure 1a). In contrast, BVOCs displayed an opposite seasonal pattern, with concentrations peaking in summer.

OVOCs were the dominant group across all seasons, accounting for 72%-77% of the total concentration, with $C_xH_yO_1$ and $C_xH_yO_2$ accounting for 51%-53% and 17%-24%, respectively (Figure 1b). $CH_4O$ was the most abundant OVOC species, with average concentration ranging from 5.98-10.10 ppbv across seasons, followed by $C_2H_4O_2$ (1.91-5.75 ppbv), $C_3H_6O$ (4.22-5.67 ppbv) and $C_2H_4O$ (1.85-3.86 ppbv), as shown in Figure S2. A statistical summary of

VOC/OVOC concentrations for each season is provided in Table S2.

The diurnal variations of OVOC subgroups across three seasons are shown in Figure 1c. $C_xH_yO_{1-3}$ groups displayed similar diurnal patterns in different seasons, characterized by pronounced daytime enhancements, particularly for species such as $C_2H_4O$, $C_3H_6O$, $C_4H_6O$ and $C_2H_4O_2$ (Figure S3). These species increased at about 7:00 local time (LT), peaked during

12:00-17:00 LT in the afternoon, and then gradually decreased, which was aligned closely with the diurnal patterns of $O_3$ (Figure S4), indicating their likely formation through photochemical reactions. Notably, $C_4H_6O$ (methyl vinyl ketone and methacrolein), the key oxidation products of isoprene (Li et al., 2021), showed pronounced photochemical daytime peaks but significantly higher concentrations in summer compared to autumn and early winter, consistent

with the seasonal trend of its precursor (Figure S5), which was different from other OVOCs formed by anthropogenic precursors. $CH_4O$ exhibited clear daytime enhancements in summer and autumn but showed no distinct diurnal pattern in early winter (Figure S3). Instead, it maintained relatively high levels throughout the day and night in early winter, likely reflecting the larger contributions from regional background sources, such as anthropogenic emissions

and background transport (Brito et al., 2015; Huang et al., 2019). These findings underscore



the significance of OVOCs in atmospheric chemistry, given their abundance and complex roles in photochemical reactions.

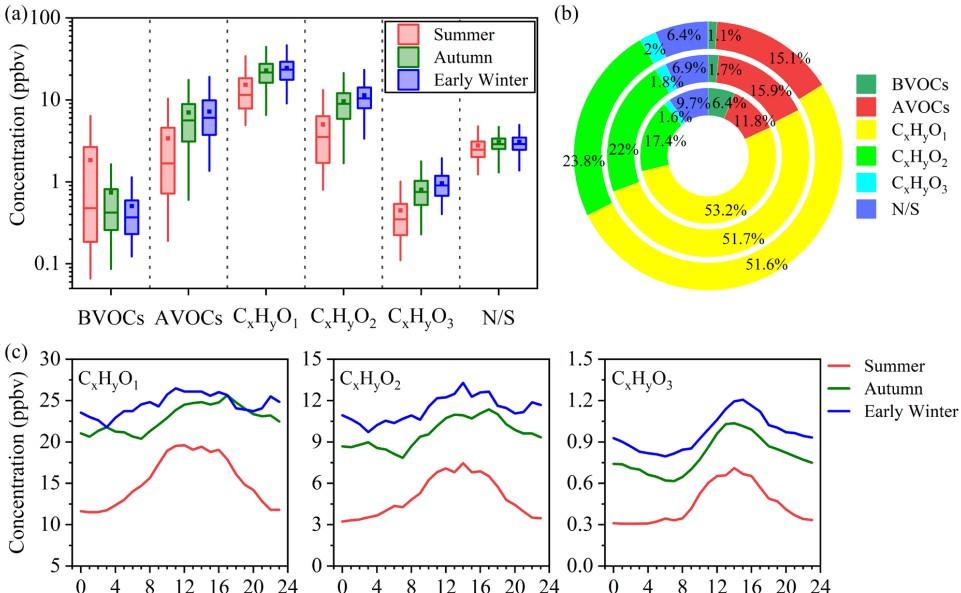

Figure 1. (a) The concentrations of different VOC groups in summer, autumn and early winter. The box plots show mean values (square), median (line within the box), interquartile range (IQR, 25%-75%), and whiskers extending to ±1.5 × IQR. (b) The contributions of different VOC groups to total concentration in summer (inner), autumn (middle), and early winter (outer). (c) Diurnal variations of OVOC subgroups across three seasons.

### 3.2 O₃-precursor relationships

To further evaluate the contribution of various VOCs to $O_3$ formation, the RIR values of key $O_3$ precursors (including BVOCs, AVOCs, OVOCs, and $NO_x$) were calculated for different seasons based on the model simulations. It should be noted that the subgroups of OVOCs and AVOCs analyzed here differ slightly from those in Section 3.1, due to the limitations that MCM does not include all mechanisms for all observed OVOC species. Additionally, $C_2$-$C_{10}$ hydrocarbons measure by GC-MS/FID/ECD were also included in the subgroup of AVOCs for this analysis. The species included in RIR calculation comprised 3 BVOC species, 45 AVOC species and 63 OVOC species, with detailed information summarized in Table S4.

The RIR values for all VOCs subgroups and $NO_x$ were positive across three seasons (Figure 2a), indicating a transition regime of $O_3$ formation in the study region and that reduction





in VOCs and/or NO$_x$ would lead to decreases in O$_3$ levels. This result was different from previous studies which reported dominantly VOC-limited regime of O$_3$ formation in Hong Kong (Liu et al., 2021; Zhang et al., 2007; Cheng et al., 2010; Guo et al., 2013). In summer,

NO$_x$ exhibited the highest RIR value (0.65), followed by BVOCs (0.21) and OVOCs (0.16). Similar trend was observed during high-O$_3$ episode days (Figure S6), indicating that summer O$_3$ formation is more sensitive to NO$_x$. By contrast, in autumn and early winter, OVOCs exhibited the highest RIR values (0.42-0.48), followed by NO$_x$ (0.25-0.35) and BVOCs (0.14-0.16), indicating that O$_3$ formation in these seasons is more sensitive to OVOCs. The dramatic

seasonal increase in the RIR value of OVOCs, coupled with a decline in those of NO$_x$ highlights the need for different control strategies to effectively reduce O$_3$ levels depending on the seasons. Previous RIR studies of O$_3$ formation have primarily focused on AVOCs, BVOCs and NO$_x$, with limited consideration of OVOCs (Wang et al., 2018b; Tan et al., 2018; Wang et al., 2022b; Yu et al., 2020; Zhang et al., 2008; Lin et al., 2020; Zhao et al., 2020; Guo et al., 2022). Recent

studies, however, have reported relatively high RIR values for OVOCs when they are included in the simulations, although these findings are confined to a narrow subset of OVOCs, mainly short-chain carbonyl compounds, based on low-resolution offline measurements (Shen et al., 2021; Yang et al., 2018; Wang et al., 2024; Feng et al., 2023). This study integrates a much broader spectrum of OVOCs (including aldehydes, ketones, organic acids, alcohols, phenolic

compounds, etc.) supported by high-resolution measurement into the observation-constrained modeling framework. This improved chemical comprehensiveness allows for a more robust characterization of OVOCs reactivity, particularly their contributions to radical production and O$_3$ sensitivity.

In addition, we have further examined the diurnal variation of RIR values for O$_3$

precursors in different seasons, as shown in Figure 2b. Significantly higher positive RIR values was observed for NO$_x$ than other precursors in the afternoon of summer and during the episodes (Figure S7), indicating a consistently strong sensitivity of O$_3$ production to NO$_x$. In autumn and early winter, OVOCs exhibited higher positive RIR values during the morning (9:00-11:00) and midday (11:00-14:00), underscoring their prominent role in O$_3$ formation. The majority of

NO$_x$ RIR values were positive except for the negative values in the morning (9:00-11:00), reflecting a shift from a VOC-limited regime in the morning to a transitional regime at the noon and in the afternoon. By late afternoon, the O$_3$ formation regime became increasingly sensitive to NO$_x$. Notably, in autumn, RIR values of NO$_x$ were comparable or even surpassed these of OVOCs in the afternoon. This variation is likely attributed to fresh NO$_x$ emissions from vehicle

and/or household combustion activities in the morning, which decrease in the afternoon due to



photochemical consumption, atmospheric diffusion, and dry deposition in the afternoon (Liu et al., 2021). These findings indicate a clear diurnal transformation pattern in $O_3$ formation regimes during the photochemical active seasons (autumn and early winter) in Hong Kong. Specifically, the regime transitions from VOC-limited in the morning to a transitional regime

with higher sensitivity to OVOCs at midday, and then to a general transitional regime with sensitivity to both OVOCs and $NO_x$ in the afternoon, and becomes increasingly $NO_x$ sensitive by late afternoon.

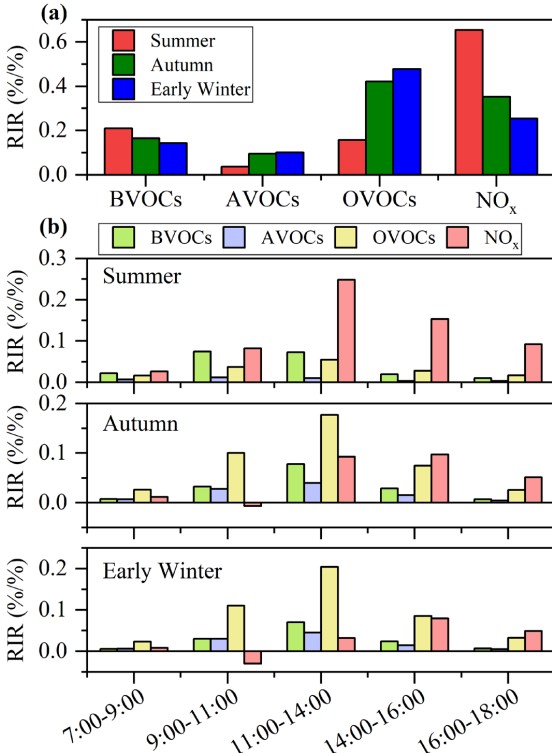

Figure 2. (a) Average RIR values of $O_3$ precursors (BVOCs, AVOCs, OVOCs, and $NO_x$)
during summer, autumn and early winter. (b) Diurnal patterns of RIR values for $O_3$ precursors across summer, autumn and early winter.

### 3.3 $O_3$ formation mechanism and radical budget

The main pathways of daytime $O_3$ production and destruction in three seasons were
further explored using the photochemical box model, as shown in Figure S8. The daytime average net $O_3$ production rate ($P_{net}$) was 5.8 ppbv/h, 6.1 ppbv/h, and 6.4 ppbv/h in summer, autumn, and early winter, respectively, much lower than that during summer high-$O_3$ episode (13.1 ppbv/h, Figure S8b). These trends were consistent with the diurnal patterns of observed



O$_3$ concentrations (Figure S9). Daytime O$_3$ production (P(O$_3$)) was dominated driven by the
reactions of HO$_2$ + NO and RO$_2$ + NO, contributing 43.8%-53.0% and 47.0%-56.2%,
respectively. Among the diverse RO$_2$ + NO reactions (involving over 1000 different RO$_2$
radicals), the top 10 pathways contributed 51.1%-54.3% of the total production rates (Figure
S10). CH$_3$O$_2$ + NO was the dominant pathway, accounting for 13.5%-19.3% of O$_3$ production.
The CH$_3$O$_2$ radical could be generated through various reactions, including the photolysis of
OVOCs (e.g., acetaldehyde, acetone) and VOCs reactions with OH radical (e.g., acetic acid),
etc. Additionally, reactions of C$_2$ radicals with NO contributed significantly (10.4%-18.5%),
with CH$_3$CO$_3$ + NO being the most prominent (10.4%-15.2%). Seasonal variations in RO$_2$ +
NO pathways were evident. RO$_2$ radicals from BVOCs (such as isoprene and pinenes)
contributed more to summer (20.7%) than autumn and early winter (6.3%-6.9%), reflecting the
seasonal patterns of vegetation emissions. Conversely, RO$_2$ radicals derived from
anthropogenic sources, such as aromatics, played a larger role in autumn and early winter
(8.6%-9.1%). Daytime O$_3$ destruction also exhibited seasonal differences. In autumn and early
winter, the dominant loss pathway was OH + NO$_2$ (51.1%-57.0%), followed by VOCs + O$_3$
(17.9%-23.2%) and O$_3$ + HO$_2$ (12.6%-13.8%). In summer, however, VOCs + O$_3$ accounted for
a much higher fraction of O$_3$ loss (43.3%) compared to other seasons, especially the reactions
with BVOCs including α-pinene and isoprene.

The OH, HO$_2$, and RO$_2$ radicals play important roles in the initiation and propagation
of atmospheric photochemical reactions (Wang et al., 2022b). The daytime production budgets
of these radicals were analyzed across three seasons, as shown in Figure S11, with detailed
contributions of each pathway presented in Figure 3. Daytime OH and HO$_2$ radical production
was highest in early winter, with lower values in summer and autumn. Radical recycling via
HO$_2$ + NO was the dominant source of OH production (86.4%-94.4%) across three seasons,
while O$_3$ photolysis (via O($^1$D) + H$_2$O) served as the main primary source for OH production,
accounting for around 60% of primary OH production rate. In addition, the photolysis reactions
of OVOCs, HONO and VOCs + O$_3$ contributed 7.6%-18.3%, 2.7%-20.4%, and 9.3%-15.6%
of primary OH production rates, respectively. Similarly, the largest source of HO$_2$ radicals was
radical recycling through RO$_2$ + NO reactions, contributing 42.5%-50.2%. OVOC photolysis
accounted for 20.3%-21.0% to total HO$_2$ production rates, but dominated the primary HO$_2$
production in three seasons (95.2%-98.6%). Moreover, the reactions of OVOCs + OH (15.4%-
17.7%) and CO + OH (10.2%-18.7%) also played significantly roles in total HO$_2$ production.
The dominant source of daytime RO$_2$ production was the reaction of VOCs + OH, accounting
for 50.8%-61.1%, with OVOCs + OH contributing the majority proportion (41.2%-51.3%).



Seasonal variations in OVOCs + OH contributions aligned with observed OVOC concentrations, being highest in early winter, followed by autumn and summer. Additional $RO_2$

sources included RO decomposition (13.9%-21.3%), OVOC photolysis (10.0%-14.2%), and VOCs + $NO_3$ (9.6%-14.1%). OVOC photolysis was also an important primary source of daytime $RO_2$ production (36.0%-57.0%). During summer high-$O_3$ episodes, daytime $RO_x$ radical production rates were significantly higher, and $O_3$ photolysis and VOCs + $O_3$ reactions contributed more substantially to $RO_x$ production (Figure S12). Furthermore, the results

highlight the importance of VOCs + $NO_3$, especially OVOCs + $NO_3$, as a source of $RO_2$ radicals in the daytime photochemistry, particularly in polluted atmosphere, consistent with prior observations at an urban site in Hong Kong (Xue et al., 2016). In total, OVOCs played a significant role in the formation of $HO_2$ and $RO_2$ radicals across all three seasons. OVOC-related reactions, including OVOC photolysis, OVOCs + OH oxidation, and OVOCs + $NO_3$,

accounted for 36.4%-38.5% and 59.1%-73.4% of daytime $HO_2$ and $RO_2$ production, respectively. These results emphasize the critical importance of OVOCs in sustaining radical cycling and driving photochemical $O_3$ formation, especially in urban and semi-urban atmospheric environments.

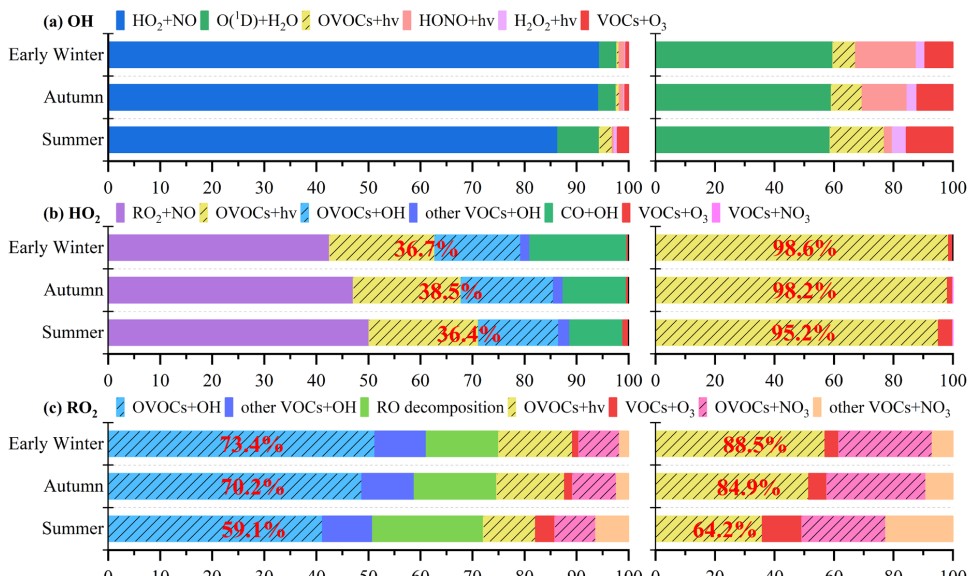

Figure 3. Contributions of key pathways for total daytime production rates (left) and primary daytime production rates (right) of (a) OH, (b) $HO_2$ and (c) $RO_2$ radicals across seasons. OVOC-related reactions including OVOC photolysis (OVOCs + hv), OVOCs oxidation by OH radicals (OVOCs + OH), and OVOCs oxidation by $NO_3$ (OVOCs + $NO_3$) are shaded by oblique lines. The percentage in red represents the contribution of OVOC-related reactions for overall

(left) and primary (right) daytime production rates of $HO_2$ and $RO_2$ radicals.



### 3.4 Importance of OVOCs in $O_3$ and radical formation

To better quantify the critical roles of OVOCs in photochemical $O_3$ and radical formation, a sensitivity simulation was conducted without constraining the observed OVOC species in the model. The comparison of observed and simulated $O_3$ concentrations under scenarios with and without OVOCs constraints across three seasons is shown in Figure S13. The simulation with OVOCs constraints successfully reproduced the observed $O_3$ concentrations in all seasons. However, without OVOCs constraints, daytime $O_3$ concentration were underestimated by 26.5% in autumn and 35.7% in early winter. The discrepancy was smaller in summer, with only minimal differences between two scenarios. This reduced sensitivity in summer is likely associated with the dominant role of $NO_x$ in $O_3$ formation in this season, coupled with elevated daytime NO concentrations, particularly during high-$O_3$ episodes. Given that $O_3$ levels were higher in early winter than other seasons and that substantial underestimations of $O_3$ were found without considering OVOCs, a detailed analysis of model performance was conducted for this period.

As shown in Figures 4a and 4b, simulated daytime $P(O_3)$ and $P_{net}$ without OVOCs constraints decreased by 44.0% and 45.1%, significantly, compared to the constrained scenario in early winter, consistent with the underestimation of $O_3$ concentration in the same period. The underestimation in $RO_2$ + NO reaction rates (45.6%) was slightly larger than that for $HO_2$ + NO (42.6%), with substantial underestimation observed for the top two $RO_2$ pathways: $CH_3O_2$ + NO (61.4%) and $CH_3CO_3$ + NO (58.6%) (Figure 4c). These discrepancies were primarily attributed to the underestimation of precursors required for $RO_2$ and $HO_2$ radical production without OVOCs constraints. Moreover, the existence of various OVOCs isomers, as detected by PTR-ToF-MS, introduces additional uncertainties in quantifying daytime $O_3$ production. Sensitivity analysis revealed that $P_{net}$ underestimations without OVOCs constraints ranged from 43.4% to 52.1%, depending on whether the minimum and maximum photolysis frequencies or $K_{OH}$ values of potential isomers were assumed (Figure 4d). The substantial underestimations of $O_3$ without OVOCs constraints underscore the critical role of OVOCs in $O_3$ formation and highlight the potentially large uncertainties in $O_3$ modeling when their contributions are inadequately represented. The differences are significantly larger than those reported in previous studies (Wang et al., 2022a; Shen et al., 2021), and likely arise from our inclusion of a wider range of OVOCs beyond the typically considered carbonyls. This enhances



the chemical completeness of the model, thereby reducing uncertainties and improving the accuracy of $O_3$ formation simulations.

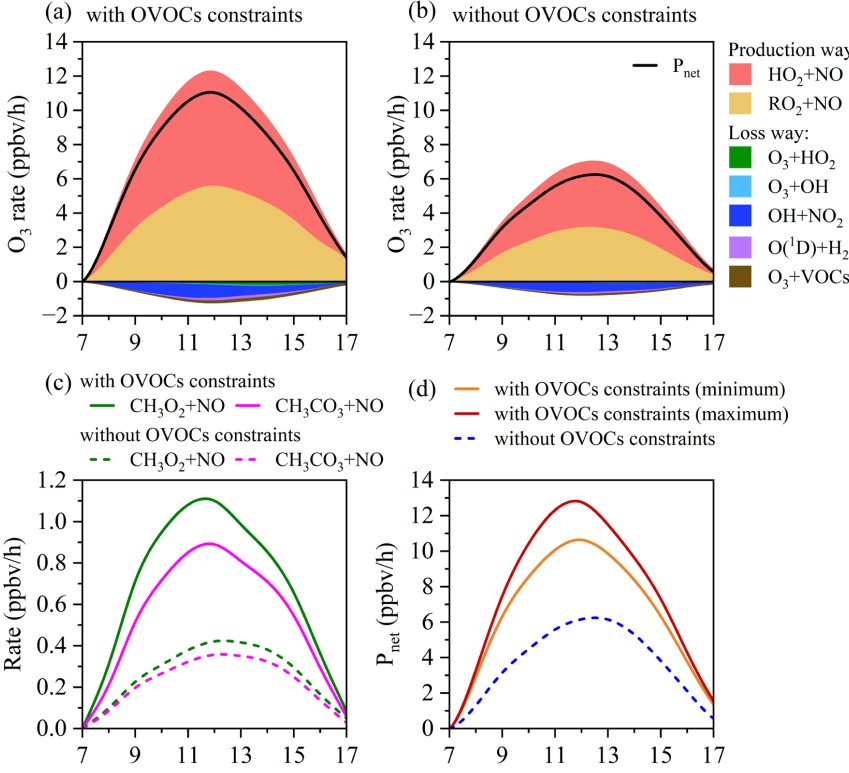

Figure 4. Model simulated daytime $O_3$ production and loss rates of main pathways in early winter (a) with and (b) without observed OVOCs constraints. The black solid line represents the net $O_3$ production rate ($P_{net}$). (c) The daytime reaction of the top two pathways of $RO_2$ + NO ($CH_3O_2$ + NO, $CH_3CO_3$ + NO) towards daytime $O_3$ formation in early winter with and without OVOCs constraints. (d) The daytime $P_{net}$ with and without observed OVOCs constraints in early winter. The orange and dark red lines represent the scenarios with the minimum and maximum OVOC contributions to $P_{net}$, respectively.

The critical role of OVOCs in modulating atmospheric radical budgets was further quantified through scenario-based simulations. As illustrated in Figure 5a, without OVOCs constraints, the production rates of OH, $HO_2$ and $RO_2$ radicals decreased significantly, with underestimations of 41.4% (40.2%-47.4% considering the minimum and maximum scenarios), 44.4% (43.2%-51.0%), and 48.0% (45.8%-57.1%), respectively. These underestimations were amplified in OVOC-related reactions, where $HO_2$ and $RO_2$ production were reduced by 58.3% (58.1%-65.5%) and 64.1% (62.0%-72.2%), respectively, underscoring the significance of OVOCs in radical cycling. These reductions were primarily attributed to the underestimation



of OVOC photolysis (37.4%-64.5%) and OVOCs + OH reactions (60.0%-71.0%) (Figure 5b). Although some carbonyl compounds are typically included in models of radical formation (Zhao et al., 2020; Yu et al., 2020; Chen et al., 2023; Han et al., 2023; Huang et al., 2020a; Liu et al., 2019; Liu et al., 2022), our results highlight that many other OVOCs, particularly those

reactive to OH oxidation and photolysis, remain overlooked, contributing to large uncertainty in simulated radical budgets. Furthermore, prominently large underestimations (up to 93.6%-95.0%) were observed for OVOCs + $NO_3$ reactions to $RO_2$ production (Figure 5b), which signals a critical gap in modeling daytime $NO_3$ oxidation chemistry of OVOCs. The significantly large underestimations of the production rates of $RO_x$ radicals and OVOC-related

reactions were mainly due to the underestimation of OVOC concentrations simulated by the photochemical model. Key OVOC species such as methanol, acetaldehyde and acetone, which are dominant and photodegradable species, were underestimated by 73%-99% in the simulations without OVOC constraints in early winter (Table S5). Similar underestimation of 10-100% of simulated OVOCs have been reported in previous studies (Wang et al., 2022a).

The missing or underestimated OVOC in these simulations may be linked to unidentified primary emission sources or unaccounted secondary sources, as current chemical mechanism, including the MCM, do not fully represent all OVOC pathways (Karl et al., 2018; Mo et al., 2016; Bloss et al., 2005). This highlights the essential role of broader OVOCs constraints in accurately representing atmospheric radical budgets and $O_3$ formation. However, even with

expanded inclusion of OVOCs in this study, certain OVOC species could not be precisely quantified, introducing residual uncertainties. Therefore, further advancements in measurement techniques are imperative to achieve more accurate OVOC quantification and reduce modeling uncertainties.





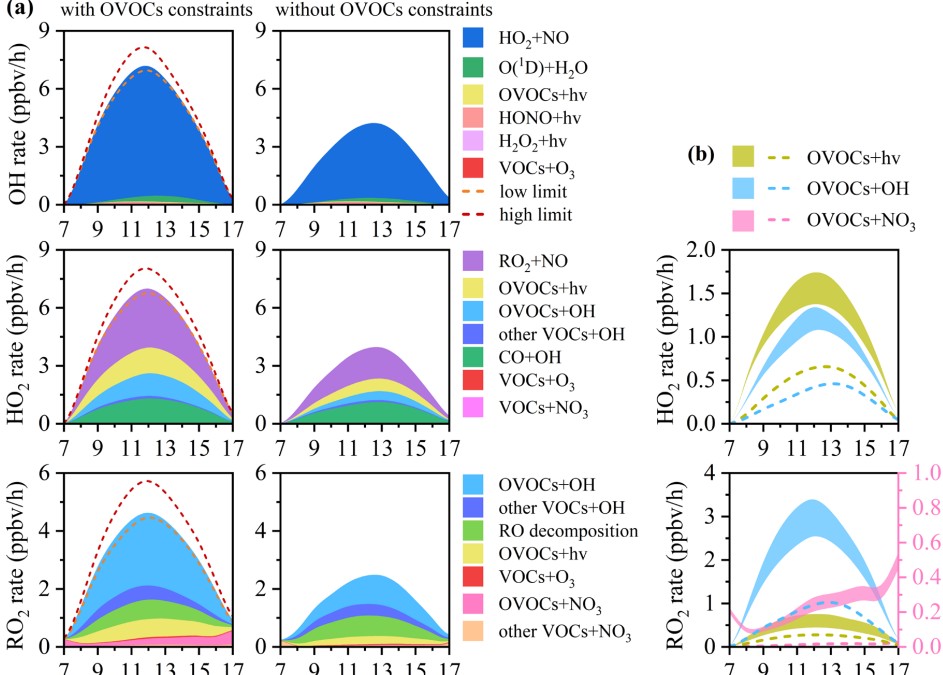

Figure 5. (a) Model simulated daytime production rates of OH, HO$_2$ and RO$_2$ radicals of main pathways in early winter with (left) and without (right) observed OVOCs constraints. The orange and dark red dash lines represent daytime production rates with minimum and maximum scenarios, respectively. (b) The reaction rates of OVOC photolysis (OVOCs + hv), OH oxidation of OVOCs (OVOCs + OH), and NO$_3$ oxidation of OVOCs (OVOCs + NO$_3$) towards daytime production of HO$_2$ and RO$_2$ radicals in early winter. The dash lines represent the scenario without OVOCs constraints. The areas represent daytime production rates with minimum and maximum scenarios.

### 3.5. Implication for O$_3$ pollution control strategies

Given the significant roles of OVOCs in O$_3$ production, EKMA O$_3$ isopleths were derived to evaluate the dependence of daytime O$_3$ production on VOCs and NO$_x$ variations. The isopleth analysis revealed a critical difference between the two scenarios: suburban Hong Kong was classified in the transition regime with OVOCs included as constraints (Figure 6a), whereas the region shifted to a VOC-limited regime without OVOCs constraints (Figure 6b). This shift highlights the importance of including OVOCs in modeling efforts, as the exclusion of OVOCs could lead to different and potentially misleading strategies for O$_3$ pollution control. Figure 6c further illustrates changes in daytime O$_3$ production in response to VOCs or NO$_x$ reductions (0% to 90%) under the two scenarios. When OVOCs were considered, O$_3$





concentration would decrease with the reduction of VOCs or NOₓ but more rapidly with VOCs,
consistent with the transition regime. In contrast, without OVOCs constraints, O₃ concentration
would initially increase with NOₓ reduction of 0%-50%, before declining at higher reductions.
Similar trends were observed in changes to daytime production rates of O₃ and ROₓ radicals,
as shown in Figure S14. Therefore, model simulation without OVOCs constraints will
overestimate the VOC-limited degree and thus overestimate the impact of VOCs reduction on
O₃ reduction, which may lead to incorrect policy implications on O₃ pollution control,
particularly on NOₓ reduction. This response suggests that the absence of OVOCs constraints
exaggerates the degree of VOC limitation, overestimating the impact of VOC reductions on O₃
control while underestimating the potential effects of NOₓ reduction.

For regions in the transition regime, such as suburban Hong Kong, simultaneous
reductions of both VOCs and NOₓ are necessary for effective O₃ control. The optimal reduction
ratios of VOCs and NOₓ, along with the response of daytime O₃ concentration, were simulated
and shown in Figure 6d. When VOCs reduction is between 0% and 40%, any reduction in NOₓ
would result in a corresponding reduction in O₃ concentration. However, when VOCs reduction
reaches 60%-90%, minor NOₓ reductions would paradoxically increase the O₃ concentration
unless NOₓ reduction is sufficiently large to outweigh the VOCs reductions. When NOₓ
reduction exceeds 90%, O₃ concentration would be reduced to below 25 ppbv, regardless of
VOC reductions. While achieving such substantial reductions in either VOCs or NOₓ emissions
presents practical challenges, a dual-focus strategy emerges as the most viable approach.
Specifically, reducing VOC emissions by 0%-40% while simultaneously minimizing NOₓ
emissions as much as possible is both pragmatic and effective. This dual strategy balances
feasibility with impact, ensuring significant O₃ reductions without introducing
counterproductive effects from disproportionate reductions in either precursor.

Although this study was conducted in a representative suburban region in Hong Kong,
the methodology and findings are broadly transferable to other urban and suburban regions
with diverse emission profiles and photochemical regimes, particularly in areas influenced by
both biogenic and anthropogenic sources. Excluding OVOCs in such environments can lead to
misclassification of O₃ formation regimes, resulting suboptimal or even counterproductive
mitigation efforts, especially where OVOCs remain undermeasured and underrepresented in
models. These insights underscore the critical need for OVOCs-inclusive modeling
frameworks to guide effective and science-based air quality management.

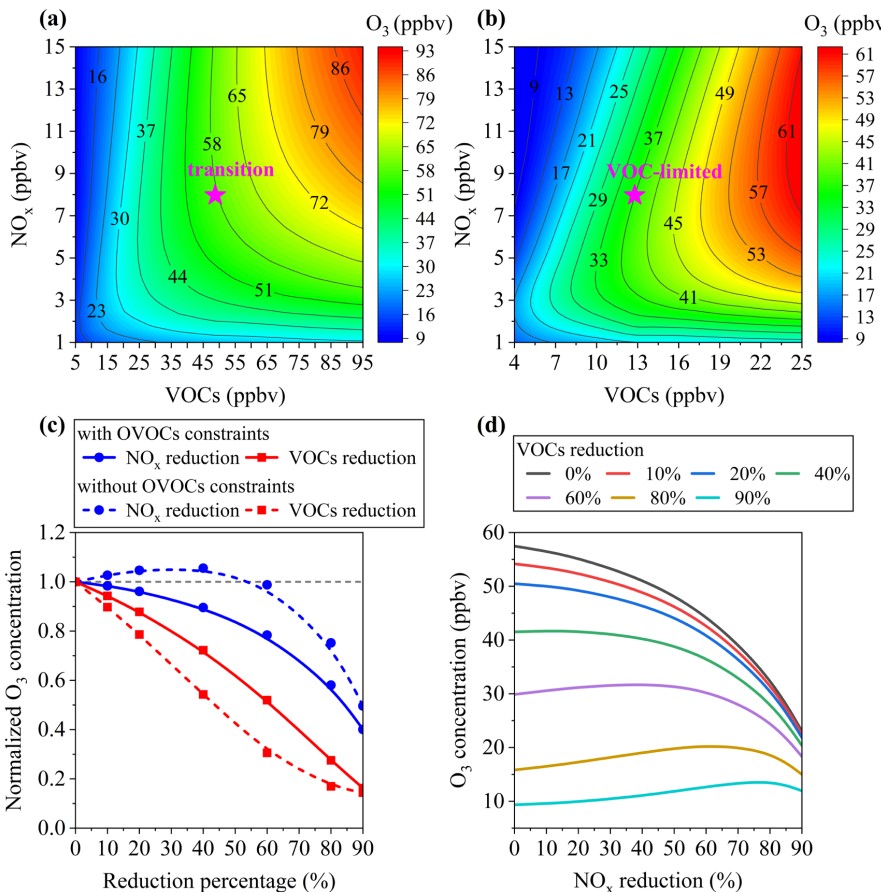

Figure 6. Isopleth diagram for average daytime O₃ production depending on NOₓ and VOCs changes in early winter (a) with and (b) without observed OVOCs constraints. The "pink star" represents the base scenario. (c) Changes in daytime O₃ production with VOCs or NOₓ reductions from 0% to 90% with and without observed OVOCs constraints. The daytime average O₃ concentrations were normalized to the corresponding values in the base run. The grey dash line represents the normalized O₃ concentration of 1.0, indicating no O₃ changing. (d) The response of daytime O₃ concentration to VOCs and NOₓ under different reduction scenarios with observed OVOCs constraints.

## 4. Conclusions

This study integrated intensive field measurements with observation-based photochemical modeling to investigate the role of OVOCs in O₃ and radical chemistry at a coastal suburban site in subtropical Hong Kong. High-resolution measurements using PTR-ToF-MS identified and quantified 117 VOC/OVOC species, among which 63 OVOCs contributed the majority (72%–77%) of total VOC concentrations across three seasons in 2021.



RIR analysis revealed a transitional $O_3$ formation regime in this suburban region, with heightened sensitivity to OVOCs, especially in autumn and early winter. Notably, $O_3$-precursor relationship also showed diurnal variations, transitioning from a VOC-limited regime in the morning to a transitional regime during midday and afternoon, underscoring the dynamic nature of $O_3$ chemistry.

Photochemical modeling demonstrated that OVOC-related reactions, including photolysis and oxidation by OH and $NO_3$ radicals, contributed substantially to radical formation, accounting for 36.4%-38.5% of daytime $HO_2$ and 59.1%-73.4% of $RO_2$ radical production. Importantly, simulations without comprehensive OVOC constraints would significantly underestimate daytime $O_3$ and $RO_x$ production rates by 41%-48% and incorrectly diagnosed the $O_3$ chemical regime. Such misclassification may lead to misguided control strategies. Compared with previous studies that only focused on a limited set of carbonyls using offline techniques, this study expands the chemical scope by including a broader suite of OVOCs through high-resolution, real-time measurements, providing a more complete assessment of OVOC-driven radical and $O_3$ formation. The mechanistic insights and modeling framework developed here offer practical value for diagnosing $O_3$ formation sensitivity and designing more effective air quality management strategies in chemically complex environments globally.

Overall, these findings underscore the critical role of OVOCs in shaping atmospheric oxidation capacity and $O_3$ formation, and highlight the need for integrating high-resolution, chemically comprehensive OVOC measurements into photochemical models. Doing so will improve the accuracy of $O_3$ formation regimes classification, reduces uncertainties in radical budgets, and supports the development of targeted, science-based, and sustainable $O_3$ pollution control strategies at both regional and global scales.

**Data availability**

The datasets associated with the current study are available from the corresponding author [z.wang@ust.hk] on reasonable request.

**Author contributions**

L.H. conducted field measurement, data analysis, model simulations and wrote the paper. Y.C. assisted in supervising the paper and provided feedback on the analysis and manuscript. D.G.



and H.S. provided VOC data measured by GC-MS/FID/ECD for model simulation. J.G. and
Y.C. provided help with model simulations. X.F., Y.X., and P.Z. provided feedback on the
analysis and manuscript. Z.W. supervised the paper and supported the funding. All the authors
participated in reviewing and editing the final version of the paper.

**Competing interests**

The authors declare that they have no conflict of interest.

**Supporting information**

The Supporting Information is available free of charge.

**Acknowledgement**

The authors would like to acknowledge the Environmental Central Facility of HKUST for
providing the air quality supersite and equipment support on ambient measurement.

**Financial support**

This study is supported by National Natural Science Foundation of China (42122062), the
Research Grants Council (RGC) of Hong Kong Special Administrative Region, China
(16209022, 16201623, 16211824), Hong Kong Environment and Conservation Fund (project
102/2023), and Guangdong Natural Science Foundation (GDST23SC13).

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
