# Peer review of "The critical role of oxygenated volatile organic compounds (OVOCs) in shaping photochemical O3 chemistry and control strategy in a subtropical coastal environment"

_EGUsphere, 2025_

## Referee Comment (RC1)

"The critical role of oxygenated volatile organic compounds (OVOCs) in shaping photochemical O3 chemistry and control strategy in a subtropical coastal environment" presents a comprehensive and well written box modelling study of the drivers of photochemical O3 formation at a subtropical coastal region in South China, particularly highlighting the importance of including a detailed representation of oxygenated VOCs (OVOCs) in the modelling study. An OVOC sensitivity study is presented, including model results with and without detailed OVOC inclusion, relevant to the broader photochemical O3 production community. I would recommend this manuscript for publication, provided a few further questions are addressed.

Main comment/suggestion:

The authors present two case studies: Box modelling with and without the inclusion of OVOCs, and the impact this has on modelling photochemical O3 production. However, in reality, it is more typical for studies to include a subset of OVOCs (though not 63, as is the case in your study), since many of these species can be measured using GC techniques alone. It would be interesting to know how good a job your box modelling does if just a "common" subset of OVOCs is incorporated (e.g. acetone, methanol, acetaldehyde). Does this subset sufficiently describe the ozone production regime? Or is it necessary to include a much more comprehensive suite of OVOCs to fully describe the chemistry? It would be worth taking a look at some literature to see which OVOCs are typically included in modelling studies for this analysis. Some examples: Whalley et al., 2018, Whalley et al., 2021, Nelson et al., 2021. In the absence of re-running the model using a hybrid between no OVOCs and 63 OVOCs, perhaps the authors already have enough information from their current model study to comment on which OVOCs are particularly important to include?

Additional comments:

Line 95 – Is it correct to say that summer is Sept 4 – Oct 12, autumn is Oct 13 – Dec 1, and winter is Dec 2 – Dec 20 in this region? This is quite a short timeframe to cover these seasons. How representative do you feel these windows are of the "summer", "autumn", and "winter" in South China?

Line 135 – We "attempted" to assign signals based on likely contributors, and that your quantification of OVOCs is semi-quantitative for uncalibrated species, is an honest and fair account of what you have done here. Whilst I completely understand that this must be the case due to instrument limitations, some readers may feel a little mis-sold up to this point, as the abstract implies that we are to expect 63 explicitly measured and quantified OVOCs. Perhaps you could be more upfront in the abstract on what you have

done to describe the OVOCs, as this is a major component of your study. Even if you just said "X quantified and Y semi-quantified OVOC species", rather than 63 quantified (abstract – line 22).

Line 160 – I was struggling to follow exactly what you've done here. "VOC species from daytime canister samples were linearly interpolated to hourly resolution for the model input". Is this just the VOCs measured using the GC-MS/FID/ECD? I assume the PTR measurements are online? Please be clear on the time resolution of the measurement of both things. How often were canister samples taken? Is interpolating the data to hourly resolution appropriate for this measurement resolution?

225 – A lot of effort has been put into speciating the OVOCs, but what about the biogenic species? The authors state there were measurements made of 2 biogenic species – isoprene, and monoterpenes. However, there are only 3 monoterpenes in the MCM (the pinenes and limonene), and they later describe their monoterpene measurement as the pinenes only. As these are highly reactive species, it is important to discuss how your assignment of the total monoterpene measurement to (what I assume is 50:50?) a- and b- pinene. There are also many monoterpene species not included in the MCM, some with faster reaction rates than these species. I understand the need for your assumption to be made, but please acknowledge the potential implications this has in the text.

300 – Just to reiterate my early point, the authors say your measurements are high-resolution here, but the resolution needs to be stated more explicitly earlier in the text.

344 - More discussion of the implications of BVOCs in your model here – again, worth pointing out that this result is based on the monoterpenes being split between the pinenes only.

397 – The authors say that including the OVOCs means that observed O3 was successfully reproduced. Are the observed and model concentrations identical? I would expect some impact from transportation to play a role here, since not all the observed O3 can be expected to be photochemically produced in situ. Please discuss.

Line 444 – "our results highlight that many other OVOCs,...,remain overlooked", and line 451 "key OVOC species such as methanol, acetaldehyde, and acetone,..., were underestimated". These key OVOCs are more typically incorporated into box modelling studies in the literature, as they can be measured using GC techniques (see my main suggestion earlier in this review).

Line 494 – When you vary the VOCs and NOx for the isopleth analysis, do you also vary OVOCs? The difficulty here is that some of your OVOCs will be formed photochemically, and some from primary sources, meaning the two will not necessarily ever decrease or increase uniformly. Do you also reduce biogenic species, or just anthropogenic? How do you navigate this issue to make the isopleth findings relevant from a policy perspective?

References:

Whalley et al (2018): Understanding in situ ozone production in the summertime through radical observations and modelling studies during the Clean air for London project (ClearfLo), Atmos. Chem. Phys., 18, 2547–2571, https://doi.org/10.5194/acp-18-2547-2018, 2018.

Whalley et al (2021).: Evaluating the sensitivity of radical chemistry and ozone formation to ambient VOCs and $NO_x$ in Beijing, Atmos. Chem. Phys., 21, 2125–2147, https://doi.org/10.5194/acp-21-2125-2021, 2021.

Nelson et al. (2021): In situ ozone production is highly sensitive to volatile organic compounds in Delhi, India, Atmos. Chem. Phys., 21, 13609–13630, https://doi.org/10.5194/acp-21-13609-2021, 2021.

---

## Referee Comment (RC2)

Review of "The critical role of oxygenated volatile organic compounds (OVOCs) in shaping photochemical O3 chemistry and control strategy in a subtropical coastal environment" by Hui et al.

The authors provide measurements of VOCs and OVOCs by PTR-MS (and GC) at a coastal urban-influenced site near Hong Kong. They use MCM box modelling to illustrate the importance of including a large range of OVOCs in models to accurately represent O3 formation. They provide quite a detailed discussion on the mechanisms through which OVOCs influence ozone formation. The paper is well written, though the discussion of the modelling could be shortened. The authors have a clear research question and use appropriate methods, but their data are not exactly suitable to provide an answer to their research question. There are large unclarities as to how the PTR-MS data was processed and how the large range of OVOCs was identified and quantified. Furthermore, instrument-specific product-ion distributions for different VOCs complicates the analysis and reduces confidence when associating an ion formula to a compound. This makes the modelling more complicated, and the conclusions of the paper may be in jeopardy. The authors should be given a chance to justify their methodology and how this affects the modelling.

The introduction reads very well.

L 94: The continuous field campaign spanned from September 4 to December 20 in 2021, covering three seasons: summer (September 4 - October 12), autumn (October 13 - December 1), and early winter (December 2 - December 20). Seasonal divisions was based on the timing of the first synoptic event, as detailed in our previous studies (Feng et al. 2023)

The categorisation of campaign dates by season seems strange. Figure S1 makes it clear though that the meteorological conditions (temperature largely) were summer-like and autumn-like during these time intervals. Maybe the authors could elaborate on their reasoning why they chose these dates, rather than just referring to Feng et al. Or include more meteorological details in the main manuscript as a figure?

L101: The sampling line was made of PEEK and the authors fitted a PTFE filter. Additionally it was heated to 80 degC.

Typically the inlet would be made of a larger and faster diameter PTFE tube, PEEK is considered stickier than PTFE. The community is split on the use of PTFE filters as to whether they affect the measurement or not. Additionally, heating the inlet to 80 degC bears the risk of evaporating organics from the aerosol phase into the gas phase. It would be important for the authors to include a short discussion here as to how their air inlet setup may have impacted the VOCs they measured.

L110: 98 Td may be considered as a low Td by some readers. Please compare to this paper: https://amt.copernicus.org/articles/12/6193/2019/amt-12-6193-2019.pdf

L121: There is a large change of transmission for compounds between m/z approx. 75 to 0. How did the authors determine the transmission of these lower mass compounds?

L125: The description of the quantification and identification of OVOCs lacks clarity. It underpins much of the modelling and the main message of the paper and needs improvement. A decision tree may be useful here.

1. Identification; Presumably the authors determined the mass of a compound by high resolution mass peak fitting. They then used an algorithm using combinations of elements to attain a mass close to the observed peak. This analysis is highly dependent on the accuracy of the mass calibration and the elements chosen and cannot differentiate isomers (e.g. types of monoterpenes). One compound might generate multiple product ions e.g. https://amt.copernicus.org/articles/18/1013/2025/ . So the approach taken by the authors may result in miss-identification. Note that product ions are highly dependent on instrument settings/operation.

2. Fragmentation; The C5H9+, commonly associated to isoprene, is also a common fragment from larger chain aldehydes e.g. https://amt.copernicus.org/articles/17/801/2024/ The PTR-MS analysis presented here does not allow to differentiate isoprene from these fragments and thus over-estimates isoprene in air measurements. These larger chain aldehydes are common emissions from cooking https://acp.copernicus.org/articles/24/4289/2024/. The authors loosely acknowledge the uncertainty related to isoprene fragmentation in Table S2, but it is unclear how this uncertainty is propagated into the modelling. The benzenoids also suffer from fragmentation and the PTR-MS analysis presented here does not allow to distinguish primary ion from fragments or the degree of fragmentation which is crucial for quantification.

3. Quantification; The authors apply a kPTR of $2 \times 10^{-9}$ cm3 s−1 for all non-calibrated compounds. Note they only calibrate for 19 VOC/OVOCs. A more accurate quantification could be achieved by using molecule specific kPTR rates e.g. https://www.sciencedirect.com/science/article/pii/S1387380616302494 and others. The authors should make it clear how much of the carbon mass or how many of the compounds (CxHyO1-3) is quantified in different ways using direct calibrations, molecule-specific kPTR or generic kPTR of $2 \times 10^{-9}$ cm3 s−1.

The authors acknowledge some uncertainty associated with this analysis, but the way this is dealt with is not clear or sufficient. E.g. "Most molecular formulas were therefore evenly distributed among potential isomers (e.g., phenols, nitrophenols)" – Conceptually this is incorrect as the isomer composition varies largely amongst different air masses and emission sources. This issue applies also to the monoterpenes. L135 to 142 reads well.

The reviewer acknowledges that the main compounds of interest may be quantified satisfactorily here for use in MCM modelling. If so, the authors should make it clear that the compounds they identified and quantified with high confidence are the ones important for O3 formation. If it turns out that the poorly quantified compounds are important for O3 formation, the conclusions from the paper are not as convincing.

It is not clear how the authors handled primary emissions e.g. isoprene and the oxidation products (MVK etc.) in the model and measurement. If the modelled and measured MVK etc. concentrations are similar, it increases the confidence that the isoprene measurements are not overly influenced by large chain aldehyde fragments.

L143; Have the authors considered comparing the GC measurements to the continuous PTR measurements to increase the confidence in their measurement e.g. for isoprene, monoterpenes, benzenoids etc.

L161; "VOC species from daytime canister samples were linearly interpolated to hourly resolution for the model input (Yang et al., 2018)" "These approximations were used primarily to

pre-run the model and were not expected to affect daytime simulation results (Chen et al., 2020)" – Can the authors provide more details how this was done please and how this could affect the conclusions from the paper?

L243 and onwards: Give the compound names these formulas have been identified as please.

L246; Could boundary layer height be influencing the diurnal variations?

L285: Given isoprene and monoterpenes show such a high RIR, uncertainties from their quantification/identification need to be assessed. See previous comment.

L293; "with limited consideration of OVOCs" – I agree with the other reviewer; Previous studies have included key OVOC species e.g. methanol, acetone and acetaldehyde. The novelty of this work lies in the inclusion of OVOCs which were previously not included and semi-quantified here with PTR-ToF.

Sections 3.2 and 3.3 read really well.

L415; "Sensitivity analysis revealed that Pnet underestimations without OVOCs constraints ranged from 43.4% to 52.1%, depending on whether the minimum and maximum photolysis frequencies or KOH values of potential isomers were assumed (Figure 4d)" It's great to have this sensitivity analysis, but the authors could have provided it at the beginning of the model discussion maybe? More technical details on the upper and lower bound constraints used in the modelling should be provided.

Sections 3.4 and 3.5 should be substantially shortened (e.g. by 30 %) as much if this feels like repetition.

The conclusion reads well

Data availability: Emailing the authors to obtain the data is not adequate, nor compliant with publisher's guidance. Please upload on a repository.

---

## Author Comment (AC1)

**Response to Referee #1**

Thanks very much for your time reviewing our manuscript entitled "The critical role of oxygenated volatile organic compounds (OVOCs) in shaping photochemical O3 chemistry and control strategy in a subtropical coastal environment". We are very grateful to the reviewers for their valuable and helpful suggestions for our manuscript. We have made all the suggested changes and clarifications. We believe that the manuscript has been significantly improved based on those suggestions.

Our point-by-point responses to reviewers' comments are as follows. We repeat the comments raised by the reviewers in *black italic* font and give our replies in the indent and normal font, and with the revised text in blue. The line numbers mentioned in responses correspond to the revised manuscript.

**Evaluation:** The authors provide measurements of VOCs and OVOCs by PTR-MS (and GC) at a coastal urban-influenced site near Hong Kong. They use MCM box modelling to illustrate the importance of including a large range of OVOCs in models to accurately represent O3 formation. They provide quite a detailed discussion on the mechanisms through which OVOCs influence ozone formation. The paper is well written, though the discussion of the modelling could be shortened. The authors have a clear research question and use appropriate methods, but their data are not exactly suitable to provide an answer to their research question. There are large unclarities as to how the PTR-MS data was processed and how the large range of OVOCs was identified and quantified. Furthermore, instrument-specific product-ion distributions for different VOCs complicates the analysis and reduces confidence when associating an ion formula to a compound. This makes the modelling more complicated, and the conclusions of the paper may be in jeopardy. The authors should be given a chance to justify their methodology and how this affects the modelling.

**Response:** We appreciate the comments and constructive suggestions from the reviewer. We have revised the manuscript and addressed the issues according to the reviewer's comments.

**1:** *The introduction reads very well.*

**Response:** We sincerely thank the reviewer for the positive feedback on the introduction.

**2:** L 94: The continuous field campaign spanned from September 4 to December 20 in 2021, covering three seasons: summer (September 4 - October 12), autumn (October 13 – December 1), and early winter (December 2 - December 20). Seasonal divisions was based on the timing of the first synoptic event, as detailed in our previous studies (Feng et al. 2023).

The categorisation of campaign dates by season seems strange. Figure S1 makes it clear though that the meteorological conditions (temperature largely) were summer-like and autumn-like during these time intervals. Maybe the authors could elaborate on their reasoning why they chose these dates, rather than just referring to Feng et al. Or include more meteorological details in the main manuscript as a figure?

**Response:** We appreciate the reviewer's comment. We added a figure of meteorological parameters including upper-level wind direction, sea level pressure and dew point and more discussion on seasonal classification during the measurement period. Since Hong Kong is located in a subtropical region where seasonal weather patterns are influenced by the Asian monsoon system. As a result, the direction of upper-level winds is commonly used to characterize seasonal transitions in this region. In this study, the seasonal classification follows

the approach adopted in previous studies (Feng et al., 2023), which is based on the occurrence of synoptic-scale events and abrupt changes in key meteorological parameters, including upper-level wind direction, sea-level pressure, and dew point temperature. The temporal variation of upper-level wind direction, sea level pressure and dew point in Hong Kong measured by Hong Kong Observatory Station from July 2021 to March 2022 was shown in Figure S1. A sudden increase in sea-level pressure accompanied by a notable decrease in dew point on 12 Oct 2021 indicated intrusion of relatively cold air masses, making the transition from summer to autumn. During the summer period, upper-level winds were predominantly easterly and/or southeasterly, reflecting typical monsoonal circulation patterns (Wong et al., 2022). Similarly, the transition from autumn to winter was characterized by abrupt changes in both dew point and upper-level wind direction, with the characteristics of cold high-pressure climate. Moreover, in winter, upper-level winds were primarily westerly and/or northwestly, consistent with the influence of the East Asian winter monsoon (Wong et al., 2022; Li et al., 2016). Therefore, the seasonal classification and selected period in this study is considered representative of the characteristic features of summer, autumn, and early winter.

**The revised text reads:**

**L96-L99**: Seasonal classification in this study was based on the occurrence of synoptic events and abrupt changes in key meteorological parameters, including upper-level wind direction, sea-level pressure, and dew point temperature (Figure S1), as detailed in our previous studies (Feng et al., 2023).

Figure S1. Temporal variation of upper-level wind direction, sea level pressure and dew point in Hong Kong measured by Hong Kong Observatory Station from July 2021 to March 2022. The seasonal transition from summer to early winter was characterized by a rapid shift from high dew point and low sea-level pressure to cold and high-pressure systems, accompanied by a change in upper-level wind direction from easterly/southeasterly to westerly/northwesterly (Li et al., 2016; Wong et al., 2022).

**3:** L101: The sampling line was made of PEEK and the authors fitted a PTFE filter. Additionally it was heated to 80 degC. Typically the inlet would be made of a larger and faster diameter PTFE tube, PEEK is considered stickier than PTFE. The community is split on the use of PTFE filters as to whether they affect the measurement or not. Additionally, heating the inlet to 80 degC bears the risk of evaporating organics from the aerosol phase into the gas phase. It would be important for the authors to include a short discussion here as to how their air inlet setup may have impacted the VOCs they measured.

Response: Thank you for your insightful comment regarding the sampling inlet setup. We apologize for the lack of clarity in our original description. In fact, ambient air was drawn into from a 1/4-inch stainless-steel sampling manifold with an inert silicon-based coating at a flow rate of 5 L min-1. Following this section, a PTFE membrane particle filter was installed to prevent particulate matter, dust and debris from entering the instrument. However, as the reviewer rightly pointed out, PTFE membrane filters may cause slight adsorption of certain highly polar or low-volatility organic compounds. To minimize this potential cumulative effect, we replaced the filter frequently during the sampling campaign. Downstream of the PTFE filter, a subsample of filtered air via a 1/16-inch PEEK tubing was directed to the inlet of the PTR at a frow rate of 100 mL min-1. As rightly pointed out, PEEK is known to have relatively higher surface adsorption compared to PTFE. To mitigate this, and to reduce humidity-related effects and wall losses, the PEEK tubing immediately upstream of the drift tube was heated to 80 °C. This temperature was chosen as it provides a balance between maintaining the thermal stability of gas-phase target compounds and minimizing condensation or adsorption prior to ionization and detection. Of course, as the reviewer correctly noted, heating the inlet line to 80 °C may have the unintended effect of volatilizing a small fraction of organic compounds from small droplets or low-concentration aerosol particles, if present in the sampled air. In such cases, these compounds could enter the gas phase and be detected by the PTR-ToF-MS. We agree that this potential influence should be clearly acknowledged in the manuscript, and we will revise the text accordingly to reflect this consideration.

**The revised text reads:**

L105-L115: Ambient air was drawn from a 1/4-inch stainless-steel sampling manifold with an inert silicon-based coating at a flow rate of 5 L min-1, and a subsample of filtered air via a 1/16-inch polyetheretherketone (PEEK) tube was directed to the PTR at a frow rate of 100 mL min-1. A polytetrafluoroethylene (PTFE) membrane particle filter was installed to prevent particulate matter, dust and debris from entering the instrument. To minimize potential cumulative adsorption effects, the filter was replaced frequently throughout the campaign. The sampling inlet was maintained at 80 °C throughout the measurements to mitigate humidity-related effects, reduce adsorption losses, and ensure gas-phase stability of target compounds prior to ionization and detection. It should be noted that this heating may unintentionally promote the volatilization of some organic compounds from aerosols, thus causing positive artifacts.

**4:** L110: 98 Td may be considered as a low Td by some readers. Please compare to this paper: https://amt.copernicus.org/articles/12/6193/2019/amt-12-6193-2019.pdf

Response: We appreciate the reviewer's insightful comment and the suggestion to consult the paper by Holzinger et al. (2019). We have carefully reviewed the recommended literature and conducted a broader survey of the E/N ratios commonly used in PTR-MS applications. Currently, E/N settings in PTR-MS typically range from 80 to 120 Td, with 100-120 Td commonly used in many studies. Lower E/N values help reduce fragmentation but may increase the formation of water hydronium clusters, leading to greater humidity dependence. In contrast, higher E/N values can cause extensive fragmentation, complicating VOC identification and quantification. Holzinger et al. (2019) systematically compared instrument responses across multiple PTR-MS systems under various operating conditions, covering an E/N range of 60-170 Td. They reported that at E/N < 100 Td (60-90 Td, particularly at 60 Td), the primary ion distribution is dominated by water hydronium clusters, which can interfere with accurate VOC quantification. Notably, at 80 Td, the TOF 1000 Ultra—the same model used in our study—produced a higher proportion of H3O+ compared to other instruments.

Moreover, the "high E/N" range discussed in that study primarily spans 100-135 Td. In this context, our operating E/N of 98 Td is quite close to the high E/N range. Nevertheless, we acknowledge that our original manuscript may have inaccurately referred to 98 Td as a "relatively high E/N" setting. We have revised and added clarification in the updated manuscript.

The revised text reads:

**L118-L122**: Lower E/N ratios can lead to a higher proportion of primary ions forming water hydronium clusters (Holzinger et al., 2019), and thus the E/N of 98 Td was selected to balance ion fragmentation and water cluster formation, which can effectively suppress water clusters formation thereby minimize the strong humidity dependence of the target species (Yuan et al., 2017).

**5:** L121: There is a large change of transmission for compounds between m/z approx. 75 to 0. How did the authors determine the transmission of these lower mass compounds?

Response: We appreciate the reviewer's question regarding transmission efficiency for lowmass compounds. In our campaign, transmission correction was initially performed using calibration gas standards with well-known proton transfer reaction rates, including benzene (m/z 79.054), toluene (m/z 93.070), m-xylene (m/z 107.086), 1,2,4-trimethylbenzene (m/z 107.086)121.101), dichlorobenzene (m/z 146.976), and trichlorobenzene (m/z 180.937). Those species are commonly used in previous studies (Sarkar et al., 2016; Zhou et al., 2019; Zhang et al., 2022). In addition,  $H_3^{18}O^+$  (m/z 21.022), naturally present in the PTR-MS spectrum, was included as a reference point at the low-mass end to help constrain the empirical transmission correction curve. We acknowledge that the absence of calibrated standards below m/z 75 may introduce additional uncertainty in the transmission correction for low-mass species. However, we have conducted sensitivity calibration and verified the transmission with some low mass compounds, such as methanol (m/z 33.034), acetonitrile (m/z 42.034), acetone (m/z 59.049), isoprene (m/z 69.07), as details in Table S1. These tests confirmed the reliability of the derived transmission correction in the low-mass region. In future work, we plan to expand the range of low-mass VOC standards used in calibration to further enhance the accuracy of transmission correction across the full m/z spectrum.

- **6:** L125: The description of the quantification and identification of OVOCs lacks clarity. It underpins much of the modelling and the main message of the paper and needs improvement. A decision tree may be useful here.
- 1. Identification; Presumably the authors determined the mass of a compound by high resolution mass peak fitting. They then used an algorithm using combinations of elements to attain a mass close to the observed peak. This analysis is highly dependent on the accuracy of the mass calibration and the elements chosen and cannot differentiate isomers (e.g. types of monoterpenes). One compound might generate multiple product ions e.g. https://amt.copernicus.org/articles/18/1013/2025/. So the approach taken by the authors may result in miss-identification. Note that product ions are highly dependent on instrument settings/operation.
- 2. Fragmentation; The C5H9+, commonly associated to isoprene, is also a common fragment from larger chain aldehydes e.g.

https://amt.copernicus.org/articles/17/801/2024/ The PTR-MS analysis presented here does not allow to differentiate isoprene from these fragments and thus over-estimates isoprene in air

measurements. These larger chain aldehydes are common emissions from cooking https://acp.copernicus.org/articles/24/4289/2024/. The authors loosely acknowledge the uncertainty related to isoprene fragmentation in Table S2, but it is unclear how this uncertainty is propagated into the modelling. The benzenoids also suffer from fragmentation and the PTR-MS analysis presented here does not allow to distinguish primary ion from fragments or the degree of fragmentation which is crucial for quantification.

3. Quantification; The authors apply a kPTR of  $2 \times 10^{-9}$  cm3 s-1 for all non-calibrated compounds. Note they only calibrate for 19 VOC/OVOCs. A more accurate quantification could achieved by using molecule specific *kPTR* rates https://www.sciencedirect.com/science/article/pii/S1387380616302494 and others. The authors should make it clear how much of the carbon mass or how many of the compounds (CxHyO1-3) is quantified in different ways using direct calibrations, molecule-specific kPTR or generic kPTR of  $2 \times 10-9$  cm3 s-1.

**Response:** We fully agree that the identification and quantification of VOCs/OVOCs using PTR-ToF-MS are fundamental to the interpretation of our measurements and the robustness of the conclusions. In response to the reviewer's suggestions, we have revised the manuscript substantially to enhance the clarity and scientific rigor of our approach to VOC identification and quantification.

**Identification:**

In this study, the identification of VOCs/OVOCs was primarily based on molecular mass, elemental combinations consisting of carbon, hydrogen, oxygen, and nitrogen atoms, reflecting plausible atmospheric molecules and functional groups. In addition, identification was also informed by previously reported high-resolution mass spectrometry results. We acknowledge that this approach cannot resolve structural isomers and may introduce some uncertainties. To address this, we applied additional consideration during the identification process. For example, as highlighted by the reviewer and mentioned literature, some ions measured by PTR can be affected by water cluster interferences, which increase the uncertainty in compound attribution. For example, the ion C4H6H+ (m/z 55.054) is susceptible to interference from H3O+(H2O)2 water cluster, which was therefore excluded from identification and subsequent analysis in this study.

**Fragmentation:**

We fully agree with the reviewer that fragmentation is also an important factor impacting compound identification in PTR-MS measurements. As reported by Coggon et al. (2024), long-chain aldehydes such as nonanal and octanal can fragment in the PTR-MS to produce C5H9+ (m/z 69), potentially leading to the misidentification of isoprene. Similarly, for benzenoid compounds, the C6H7+ ion (m/z 79), commonly attributed to benzene, may be influenced by fragmentation of ethylbenzene, thereby complicating accurate source attribution. Therefore, assumptions regarding the distribution of isomers and potential fragment interferences were made based on previous studies using the gas chromatography pre-separation. For example, for the attribution of C5H8 signals, isoprene was allocated the fraction of 63% reported in previous studies employing PTR-MS measurements coupled with GC, which effectively minimizes interference from fragments of higher molecular compounds (Koss et al., 2018).

**Quantification:**

In this study, the quantification of VOCs/OVOCs measured by PTR-MS was performed using a combination of calibration and theoretical estimation. Specifically, 18 compounds were calibrated using multi-component VOC gas standards delivered via a Liquid Calibration Unit. For the remaining 98 VOCs lacking available calibration standards, concentrations were

estimated using an assumed proton-transfer reaction rate coefficient of  $2 \times 10^{-9}$  cm3 s-1, along with a mass-dependent transmission correction following the method described by Zhang et al. (2022). We acknowledge that this approach still have uncertainties for uncalibrated compounds. The method referenced by the reviewer, which estimates proton-transfer rate constants (k values) based on molecular properties such as polarizability and dipole moment, is scientifically robust, broadly applicable, and offers a promising pathway to reduce such uncertainties (Sekimoto et al., 2017). We recognize the value of this approach and plan to adopt it in future work to estimate k values for VOCs measured by PTR-MS. This will enable improved estimates of instrument sensitivity and VOC concentrations, thereby enhancing the overall accuracy of PTR-based quantification.

**The revised text reads:**

**L136-L146**: This attribution was based on molecular mass, elemental combinations consisting of carbon, hydrogen, oxygen, and nitrogen atoms, reflecting plausible atmospheric molecules and functional groups, as well as prior studies utilizing high-resolution mass spectrometry (Yuan et al., 2017; Koss et al., 2018; Wu et al., 2020), as summarized in Table S2. For the remaining 98 VOC/OVOC species lacking available calibration standards, concentrations were determined using an assumed proton transfer reaction rate coefficient of  $2 \times 10^{-9}$  cm3 s-1, combined with mass-dependent transmission correction (Zhang et al., 2022). To reduce uncertainties for uncalibrated compounds, an approach developed by Sekimoto et al. (2017), which estimates proton-transfer rate constants based on molecular properties such as polarizability and dipole moment, provides a scientifically robust method that could be applied in future work.

L166-L174: To minimize uncertainties arising from water cluster interferences, ions susceptible to such effects, such as C4H6H+ (m/z 55.054), which overlaps with the H3O+(H2O)2 cluster, were excluded from compound identification and subsequent analysis in this study. Regarding impacts from fragmentation, for example, C5H8 may be affected by fragment interferences from higher-carbon aldehydes and cycloalkanes (Coggon et al., 2024; Claflin et al., 2021; Yuan et al., 2017; Zhang et al., 2025), therefore, the attribution of C5H8 to isoprene follows the proportion of 63% reported in previous studies employing PTR-MS coupled with GC pre-separation, which effectively minimizes interference from fragments of higher molecular compounds (Koss et al., 2018).

7: The authors acknowledge some uncertainty associated with this analysis, but the way this is dealt with is not clear or sufficient. E.g. "Most molecular formulas were therefore evenly distributed among potential isomers (e.g., phenols, nitrophenols)" — Conceptually this is incorrect as the isomer composition varies largely amongst different air masses and emission sources. This issue applies also to the monoterpenes. L135 to 142 reads well.

Response: We agree with the reviewer that the distribution of structural isomers can vary substantially across different air masses and emission sources. However, for the majority of compounds in our dataset, there is currently no reliable or systematic method to resolve and assign individual isomers. As such, we adopted an equal distribution approach for these compounds. For specific compounds—particularly many aldehydes and ketones—which have been identified based on prior studies using high-resolution mass spectrometry coupled with gas chromatography. We acknowledge that although we attempted to assign signals based on likely contributors informed by literature, this approach introduces uncertainties in the molecular-level identification due to variability in instrument sensitivity, resolution, ambient conditions, and sampling periods across studies. These factors can affect the observed chemical

composition and relative contributions of individual species, thereby influencing the accuracy of signal attribution and subsequent model inputs.

The revised text reads:

L148-L163: PTR-ToF-MS is limited in its ability to differentiate between isomeric compounds, accurate quantification of individual compounds remains challenging. Most molecular formulas were therefore evenly distributed among potential isomers (e.g., phenols, nitrophenols), while specific formulas for aldehydes and ketones were identified based on prior studies using GC-PTR-ToF measurement. For example, for C10H16, given that α-pinene and βpinene are typically the predominant contributors (Kim et al., 2009; Byron et al., 2022; Kammer et al., 2020), an equal 50:50 allocation between the two species was adopted as a modeling assumption for the apportionment. As important oxidation products, C4H6O was apportioned as 48% methyl vinyl ketone (MVK), 19% methacrolein (MACR), and 33% crotonaldehyde for model simulations, based on previous studies using PTR-MS combined with GC pre-separation (Koss et al., 2018). Although we attempted to assign signals based on likely contributors informed by literature, this approach introduces uncertainties in the molecular-level identification due to variability in instrument sensitivity, resolution, ambient conditions, and sampling periods across studies. These factors can affect the observed chemical composition and relative contributions of individual species, thereby influencing the accuracy of signal attribution and subsequent model inputs.

**8:** The reviewer acknowledges that the main compounds of interest may be quantified satisfactorily here for use in MCM modelling. If so, the authors should make it clear that the compounds they identified and quantified with high confidence are the ones important for O3 formation. If it turns out that the poorly quantified compounds are important for O3 formation, the conclusions from the paper are not as convincing.

Response: We thank the reviewer for this valuable comment. We agree that the accuracy of quantified compounds is crucial for ensuring the robustness of the O3 formation analysis. In this study, due to the limited availability of calibration standards, 11 VOCs and 8 OVOC species detected by the PTR were quantitatively calibrated and thus considered high-confidence compounds. These include key aromatic hydrocarbons, isoprene, and several major OVOCs that are well recognized as important O3 precursors in previous studies. The remaining 97 species were quantified on a semi-quantitative basis using empirically derived transmission and sensitivity corrections, which we consider providing reasonable estimates of their ambient concentrations. While we acknowledge that uncertainties in PTR-based quantification for these semi-quantified species may introduce some variability in the model simulations, we have addressed this by performing sensitivity analyses to estimate the upper and lower bounds of their potential impacts on atmospheric O3 and ROx radical production. The results indicate that, despite these quantification uncertainties, the PTR-measured OVOCs still provide robust and meaningful insights into the role of OVOCs in atmospheric photochemistry and O3 formation.

**9:** It is not clear how the authors handled primary emissions e.g. isoprene and the oxidation products (MVK etc.) in the model and measurement. If the modelled and measured MVK etc. concentrations are similar, it increases the confidence that the isoprene measurements are not overly influenced by large chain aldehyde fragments.

**Response:** We appreciate the reviewer's insightful comment. In our modeling analysis, we employed the concentrations of C5H8 and C4H6O measured by PTR-ToF-MS as the input, representing isoprene and its major oxidation products (including MVK and MACR),

respectively. However, due to the inherent limitations of PTR in distinguishing structural isomers, accurate quantification of individual compounds remains challenging. To account for this, assumptions regarding the distribution of isomers and potential fragment interferences were made based on previous studies using gas chromatography pre-separation. For the attribution of C5H8H+ signals, isoprene was allocated the distribution of 63% reported in previous studies employing PTR-MS measurements coupled with GC, which effectively minimizes interference from fragments of higher molecular compounds (Koss et al., 2018). Similarly, for the C4H6OH+ signal, we assigned the contributions as 48% MVK, 19% MACR, and 33% crotonaldehyde, following established literature proportions. Regarding the comparison between modeled and observed OVOC concentrations, we acknowledge that achieving an exact match is inherently difficult. Although the MCM implemented in our model provides a detailed representation of atmospheric chemistry, it does not fully capture the formation and degradation pathways of many OVOCs, particularly long-chain compounds. Moreover, the model primarily simulates secondary chemical processes and does not include primary emissions or other complex sources, which limits its ability to accurately reproduce ambient OVOC levels. We recognize that the assumptions made for the distributions of C5H8 and C4H6O may introduce uncertainties in both quantification and model performance, and we added some discussion of associated uncertainties in the revised manuscript.

**The revised text reads:**

**L148-L149**: PTR-ToF-MS is limited in its ability to differentiate between isomeric compounds, accurate quantification of individual compounds remains challenging.

L155-L163: As important oxidation products, C4H6O was apportioned as 48% methyl vinyl ketone (MVK), 19% methacrolein (MACR), and 33% crotonaldehyde for model simulations, based on previous studies using PTR-MS combined with GC pre-separation (Koss et al., 2018). Although we attempted to assign signals based on likely contributors informed by literature, this approach introduces uncertainties in the molecular-level identification due to variability in instrument sensitivity, resolution, ambient conditions, and sampling periods across studies. These factors can affect the observed chemical composition and relative contributions of individual species, thereby influencing the accuracy of signal attribution and subsequent model inputs.

L168-L174: Regarding impacts from fragmentation, for example, C5H8 may be affected by fragment interferences from higher-carbon aldehydes and cycloalkanes (Coggon et al., 2024; Claflin et al., 2021; Yuan et al., 2017; Zhang et al., 2025), therefore, the attribution of C5H8 to isoprene follows the proportion of 63% reported in previous studies employing PTR-MS coupled with GC pre-separation, which effectively minimizes interference from fragments of higher molecular compounds (Koss et al., 2018).

**L369-L372**: Nevertheless, due to the inherent limitations of PTR-ToF-MS, accurate quantification of isomers with distinct chemical reactivities remains challenging, introducing some uncertainties in atmospheric photochemical modeling.

**10:** L143; Have the authors considered comparing the GC measurements to the continuous PTR measurements to increase the confidence in their measurement e.g. for isoprene, monoterpenes, benzenoids etc.

**Response:** Thanks for the reviewer's comment. We selected three representative compounds—C5H8, C6H6, and C7H8—which have fewer isomers in PTR measurements. These compounds correspond to isoprene, benzene, and toluene, respectively, which have been distributed based on previous studies using PTR-MS combined with GC pre-separation (Koss et al., 2018). The

comparison between PTR and GC measurements shows good agreement between PTR and GC data, with correlation coefficients (R²) of 0.70 for isoprene, 0.86 for benzene, and 0.77 for toluene. The slopes of the comparisons were 1.25, 1.63, and 1.51, respectively, indicating that PTR tended to report slightly higher concentrations than GC. This discrepancy may be attributed to contributions from structural isomers or fragments originating from higher-mass compounds, introducing some uncertainties. Nevertheless, the overall agreement between PTR and GC measurements confirms that PTR data are reliable and the deviations remain within an acceptable range. The comparison results have been included in the revised manuscript for reference.

The revised text reads:

**L182-L187**: The PTR quantified isoprene, benzene, and toluene from the signals of  $C_5H_8$ ,  $C_6H_6$ , and  $C_7H_8$  were compared with GC measurements. The two datasets showed good agreement ( $R^2 = 0.70 \sim 0.86$ ), although PTR reported slightly higher concentrations (slope =  $1.25 \sim 1.63$ ). Overall, PTR and GC measurements are generally comparable in this study, and PTR data can be considered reliable within an acceptable uncertainty range.

11: L161; "VOC species from daytime canister samples were linearly interpolated to hourly resolution for the model input (Yang et al., 2018)" "These approximations were used primarily to pre-run the model and were not expected to affect daytime simulation results (Chen et al., 2020)" — Can the authors provide more details how this was done please and how this could affect the conclusions from the paper?

**Response:** Thanks for the reviewer's comments. We added more details about the linear interpolation of canister samples and the potential impacts of such approximations for the model simulation.

The revised text reads:

**L202-L213**: For the offline canister VOC samples measured by GC-MS/FID/ECD, daytime data from 9:00 to 18:00 were linearly interpolated to an hourly resolution for the model input (Yang et al., 2018), while nighttime concentrations of unmeasured C2-C10 hydrocarbons (excluding isoprene and monoterpenes) and alkyl nitrates were estimated using linear regression relationships with continuously measured hydrocarbons (e.g., C3H6, C5H10, C6H10) and nitrophenols obtained from PTR-ToF-MS measurements. The PTR measured species used in the linear regression calculation were selected based on their strong correlations with corresponding compounds in the canister data to ensure more reliable estimates. These approximations were primarily used to pre-run the model and ensure continuous modeling, and were not expected to significantly affect the daytime simulation results, since photochemical activity is minimal during nighttime, and most hydrocarbons are less reactive in the absence of sunlight (Chen et al., 2020).

**12:** L243 and onwards: Give the compound names these formulas have been identified as please.

**Response:** Thanks for the reviewer's comment. We have added the compound names of the formulas in the manuscript.

The revised text reads:

L298-L302: CH4O (methanol) was the most abundant OVOC species, with average concentration ranging from 5.98-10.10 ppbv across seasons, followed by C2H4O2 (1.91-5.75

ppbv), C3H6O (4.22-5.67 ppbv) and C2H4O (1.85-3.86 ppbv), which primarily correspond to acetic acid, acetone and acetaldehyde, respectively.

**L305-L307**:  $C_xH_yO_{1-3}$  groups displayed similar diurnal patterns in different seasons, characterized by pronounced daytime enhancements, particularly for species such as  $C_2H_4O$  (acetaldehyde),  $C_3H_6O$  (acetone),  $C_4H_6O$  (MVK and MACR) and  $C_2H_4O_2$  (acetic acid).

L314-L316: CH4O (methanol) exhibited clear daytime enhancements in summer and autumn but showed no distinct diurnal pattern in early winter.

**13:** *L246*; *Could boundary layer height be influencing the diurnal variations?*

**Response:** Thanks for the reviewer's comment. As the reviewer suggested, the boundary layer height could influence the diurnal patterns of OVOC species, so we add relevant analysis in this section.

The revised text reads:

**L319-L321**: In addition to precursor availability and photochemical activity, diurnal variations in boundary layer height may also influence OVOC concentrations by modulating vertical mixing and accumulation processes.

**14:** *L285:* Given isoprene and monoterpenes show such a high RIR, uncertainties from their quantification/identification need to be assessed. See previous comment.

**Response:** Thanks for the valuable suggestion. We added some discussion about the quantification/identification of isoprene and monoterpenes in this section.

The revised text reads:

**L152-L155**: For example, for  $C_{10}H_{16}$ , given that  $\alpha$ -pinene and  $\beta$ -pinene are typically the predominant contributors (Kim et al., 2009; Byron et al., 2022; Kammer et al., 2020), an equal 50:50 allocation between the two species was adopted as a modeling assumption for the apportionment.

L169-L174: C5H8 may be affected by fragment interferences from higher-carbon aldehydes and cycloalkanes (Coggon et al., 2024; Claflin et al., 2021; Yuan et al., 2017; Zhang et al., 2025), therefore, the attribution of C5H8 to isoprene follows the proportion of 63% reported in previous studies employing PTR-MS coupled with GC pre-separation, which effectively minimizes interference from fragments of higher molecular compounds (Koss et al., 2018).

**L282-L285**: It should be noted that C5H8 may be affected by fragment interferences from higher-carbon aldehydes and cycloalkanes (Coggon et al., 2024; Claflin et al., 2021; Yuan et al., 2017; Zhang et al., 2025), which may potentially lead to an overestimation of BVOCs, particularly isoprene.

**L350-L355**: It should be noted that current photochemical models typically represent monoterpenes using only  $\alpha$ -and  $\beta$ -pinenes, neglecting some highly reactive species such as limonene. Moreover, gaps in the MCM, such as the absence of certain highly reactive monoterpenes and associated oxidation pathways, may further introduce uncertainties in assessing the role of BVOCs in atmospheric photochemistry.

**15:** *L293*; "with limited consideration of OVOCs" – I agree with the other reviewer; Previous studies have included key OVOC species e.g. methanol, acetone and acetaldehyde. The novelty

of this work lies in the inclusion of OVOCs which were previously not included and semiquantified here with PTR-ToF.

**Response:** We agree with the reviewer's comment. However, what we intended to emphasize is that, under the scenario without OVOC constraints, the primary reason for the underestimation of  $O_3$  and  $RO_x$  production is the substantial underestimation of OVOC concentrations in the model simulation. The reference to "methanol, acetaldehyde, and acetone being underestimated by 73%-99%" was intended merely as an example to illustrate this issue. Thus, our intention was neither to imply that these three compounds are absent from other models reported in the literature, nor to highlight their inclusion as a novel contribution of our study.

In addition, we have made another sensitivity test excluding four commonly OVOCs including acetaldehyde, acetone, MEK, and butanal based on previous studies (Whalley et al., 2021; Whalley et al., 2018; Yang et al., 2018; Feng et al., 2023; Shen et al., 2021). We then compared the production rates of O3, OH, HO2, and RO2 with those from the base case (with full OVOCs constraints) and without OVOCs constraints. The results show that excluding only these four OVOCs led to a reduction of 3.9%-9.4% in daytime O3 and ROx radical production rates, which is significantly smaller than the 41%-48% reduction observed in the case without all OVOCs constraints. This indicates that the significant underestimation of O3 and radical production rates without OVOCs constraints cannot be explained solely by common OVOC species. In other words, approximately 40% of photochemical production potential arises from OVOCs which were previously not included but semi-quantified with PTR in this study.

The revised text reads:

**L525-L528**: These discrepancies were largely due to the underestimation of multiple OVOC species, for example, methanol, acetaldehyde, and acetone were underestimated by 73%-99% in early winter simulations without OVOC constraints (Table S5).

**L517-L522**: A sensitivity test excluding four commonly OVOCs including acetaldehyde, acetone, MEK, and butanal (Whalley et al., 2021; Whalley et al., 2018; Yang et al., 2018; Feng et al., 2023; Shen et al., 2021) resulted in only a 3.9%-9.4% reduction in daytime O3 and ROx production rate, compared to a 41%-48% reduction without OVOCs constraints. This indicates that nearly 40% reduction cannot be explained by these common OVOCs alone, further highlighting the importance of comprehensive OVOCs in modeling.

**16:** Sections 3.2 and 3.3 read really well.

**Response:** We sincerely thank the reviewer for the positive feedback on these two sections.

17: L415; "Sensitivity analysis revealed that Pnet underestimations without OVOCs constraints ranged from 43.4% to 52.1%, depending on whether the minimum and maximum photolysis frequencies or KOH values of potential isomers were assumed (Figure 4d)" It's great to have this sensitivity analysis, but the authors could have provided it at the beginning of the model discussion maybe? More technical details on the upper and lower bound constraints used in the modelling should be provided.

**Response:** We thank the reviewer for positive feedback on sensitivity analysis. While we appreciate the suggestion to move the sensitivity analysis to the beginning of the model discussion, we believe it is more appropriate to retain it in its current position. The primary objective of modeling work is to investigate the role of OVOCs in atmospheric production of O3 and radicals. The sensitivity analysis, on the other hand, is intended to address the

limitations of PTR measurements—specifically, the inability to distinguish structural isomers. By applying upper and lower bound assumptions for photolysis frequencies and KOH rate constants, the analysis provides a range of possible production rates of O3 and ROx radicals, helping to assess the overall contribution of OVOCs under measurement uncertainties. And we have added more details on how the upper and lower bounds were defined in the model in Section 2.2 Photochemistry Modeling.

The revised text reads:

**L239-L248:** For OVOCs measured by the PTR with multiple isomers, each molecular formula was assigned either to the isomer with the minimum or maximum photolysis frequencies and KOH values among all plausible isomeric structures. Specifically, for OVOCs with isomers containing both aldehydes and ketones, the upper bound was defined by assigning OVOCs to aldehydes with the highest photolysis frequency, while the lower bound assumed ketones with the lowest photolysis frequency (e.g., C4H6O, C4H8O). For OVOCs whose isomers do not undergo photolysis but are susceptible to OH oxidation, the upper and lower bounds were determined based on OH reactivity, with the upper and lower bounds corresponding to the isomers with highest and lowest KOH values, respectively (e.g., C8H10O, C6H8O2).

**18:** Sections 3.4 and 3.5 should be substantially shortened (e.g. by 30 %) as much if this feels like repetition.

**Response:** Thanks for the reviewer's suggestion. We have shortened and revised Sections 3.4 and 3.5.

The revised text reads:

**Section 3.4:**

To better quantify the critical roles of OVOCs in photochemical O3 and radical formation, a sensitivity simulation was conducted without constraining the observed OVOC species in the model. The comparison of observed and simulated O3 concentrations under scenarios with and without OVOCs constraints across three seasons is shown in Figure S14. Incorporating a broader range of OVOCs improved the simulation of O3, particularly in autumn and early winter, where daytime concentrations were underestimated by 26.5% and 35.7%, respectively, without OVOCs constraints. In contrast, the discrepancy was minimal in summer, likely due to the dominant role of NOx in O3 formation and elevated daytime NO levels during high-O3 episodes. It should be noted that the model considers only in situ photochemical processes and does not include influences such as regional transport. As a result, discrepancies between observed and simulated O3 remain, especially in autumn and early winter, when periods typically influenced by the Asian monsoon, and during nighttime when photochemical activity is minimal. Given the higher observed O3 levels in early winter and the substantial underestimation without OVOC constraints, we conducted a focused evaluation of model performance for this period.

As shown in Figures 4a and 4b, simulated daytime  $P(O_3)$  and  $P_{net}$  without OVOCs constraints in early winter decreased by 44.0% and 45.1%, respectively, consistent with the underestimation of  $O_3$  concentrations during the same period. The reduction in  $RO_2 + NO$  reaction rates (45.6%) was slightly larger than that for  $HO_2 + NO$  (42.6%), with the most substantial decreases observed in  $CH_3O_2 + NO$  (61.4%) and  $CH_3CO_3 + NO$  (58.6%) pathways (Figure 4c). These reductions were primarily attributed to the underestimation of radical precursors without OVOCs constraints. Moreover, the existence of multiple OVOC isomers detected by PTR-ToF-MS, introduces additional uncertainties in quantifying daytime  $O_3$

production. Sensitivity analysis revealed that Pnet underestimations without OVOCs constraints ranged from 43.4% to 52.1%, depending on the assumed photolysis frequencies and KOH values of potential isomers (Figure 4d). These results highlight the critical role of OVOCs in O3 formation and the potentially large uncertainties in O3 modeling when their contributions are not adequately represented. The discrepancies are significantly larger than those reported in previous studies (Wang et al., 2022; Shen et al., 2021), and likely arise from our inclusion of a broader range of OVOCs beyond commonly considered carbonyls, enhancing the chemical completeness of the model and improving the simulation accuracy.

The critical role of OVOCs in modulating atmospheric radical budgets was further quantified through scenario-based simulations. As illustrated in Figure 5a, without OVOCs constraints led to significant reductions in the production rates of OH (40.2%-47.4%), HO2 (43.2%-51.0%) and RO2 (45.8%-57.1%) radicals, with ranges reflecting the minimum and maximum assumptions. These underestimations were amplified in OVOC-related reactions, where HO2 and RO2 production were reduced by 58.1%-65.5% and 62.0%-72.2%, respectively, underscoring the significance of OVOCs in radical cycling. These reductions were primarily attributed to the underestimation of OVOC photolysis (37.4%-64.5%) and OVOCs + OH reactions (60.0%-71.0%) (Figure 5b). Although some carbonyl compounds are commonly included in models of radical formation (Zhao et al., 2020; Yu et al., 2020; Chen et al., 2023; Han et al., 2023; Huang et al., 2020; Liu et al., 2019; Liu et al., 2022), our results highlight that many other photoreactive OVOCs remain overlooked, contributing to large uncertainties in radical simulations. A sensitivity test excluding four commonly OVOCs including acetaldehyde, acetone, MEK, and butanal (Whalley et al., 2021; Whalley et al., 2018; Yang et al., 2018; Feng et al., 2023; Shen et al., 2021) resulted in only a 3.9%-9.4% reduction in daytime O3 and ROx production rate, compared to a 41%-48% reduction without OVOCs constraints. This indicates that nearly 40% reduction cannot be explained by these common OVOCs alone, further highlighting the importance of comprehensive OVOCs in modeling. Furthermore, prominently large underestimations (up to 93.6%-95.0%) occurred in RO2 production from OVOCs + NO3 reactions (Figure 5b), revealing a critical gap in modeling daytime NO3 oxidation of OVOCs. These discrepancies were largely due to the underestimation of multiple OVOC species, for example, methanol, acetaldehyde, and acetone were underestimated by 73%-99% in early winter simulations without OVOC constraints (Table S5). Similar underestimation (10-100%) of simulated OVOCs have been reported in previous studies (Wang et al., 2022). Such underestimations likely stem from missing or unresolved primary and secondary OVOC sources, as current mechanisms, including the MCM, do not comprehensively represent OVOC chemistry (Karl et al., 2018; Mo et al., 2016; Bloss et al., 2005). These findings highlight the necessity of broader OVOCs constraints for accurately modeling radical budgets and O3 formation. However, residual uncertainties remain due to limitations in isomer-specific quantification, emphasizing the need for improved measurement techniques to better constrain OVOC impacts in atmospheric models.

**Section 3.5**:**

Given the critical role of OVOCs in O3 production, EKMA O3 isopleths were derived to evaluate the dependence of daytime O3 production on VOCs and NOx variations. The analysis revealed a critical difference between the two scenarios: suburban Hong Kong was classified in the transition regime with OVOCs constraints (Figure 6a), whereas it shifted to a VOC-limited regime without OVOCs constraints (Figure 6b). This highlights the importance of including OVOCs in modeling efforts, as their exclusion may lead to potentially misleading strategies for O3 pollution control. Figure 6c further illustrates changes in daytime O3 production in response to VOCs or NOx reductions (0% to 90%) under the two scenarios. With OVOCs constraints, O3 concentration would decrease with reductions in either VOCs or NOx,

but more strongly with VOCs, consistent with the transition regime. In contrast, without OVOCs constraints, O3 concentration would initially increase with NOx reduction of 0%-50%, before declining at higher reductions. Similar patterns were observed in changes to daytime production rates of O3 and ROx radicals (Figure S15). These results demonstrate that neglecting OVOCs exaggerates the VOC-limited degree and overestimate the effectiveness of VOCs reduction on O3 reduction, while underestimating the potential benefits of NOx control. This could lead to suboptimal or ineffective O3 mitigation strategies.

For regions in the transition regime, such as suburban Hong Kong, simultaneous reductions in both VOCs and NOx are necessary for effective O3 control. As shown in Figure 6d, when VOC reduction is between 0% and 40%, any reduction in NOx would result in corresponding O3 reductions. However, at VOC reductions (60%-90%), minor NOx reductions would paradoxically increase O3 levels unless NOx reduction is sufficiently large to outweigh VOC reductions. Notably, when NOx reduction exceeds 90%, O3 concentration falls below 25 ppbv, regardless of VOC levels. While such drastic emission reductions are challenging, a dual-control strategy with reduction of VOCs by 0%-40% and minimizing NOx emissions emerges as both feasible and effective, avoiding unintended increases in O3 levels. It should be noted that due to the absence of a robust mechanism to represent the nonlinear formation and diverse sources of OVOCs, we employed a simplified scaling approach based on precursor VOCs in O3 isopleth analysis. Despite inherent uncertainties, this provides a practical approximation for assessing OVOC impacts on O3 formation under varying emission scenarios.

Although this study focused on a representative suburban region in Hong Kong, the methodology and findings are broadly applicable to other urban and suburban regions with diverse emission profiles and photochemical regimes. In such environments, excluding OVOCs can lead to misclassification of O3 formation regimes, resulting in ineffective or counterproductive control strategies. These insights underscore the critical need for OVOCs-inclusive modeling frameworks to guide effective and science-based air quality management.

**19:** *The conclusion reads well.*

**Response:** We sincerely thank the reviewer for the positive feedback on the conclusion.

**20:** Data availability: Emailing the authors to obtain the data is not adequate, nor compliant with publisher's guidance. Please upload on a repository.

**Response:** Thanks for the reviewer's comment, we have uploaded the database into a repository in DataSpace@HKUST with link of https://dataspace.hkust.edu.hk/bib/ZV6FMX.

**References:**

Bloss, C., Wagner, V., Bonzanini, A., Jenkin, M. E., Wirtz, K., Martin-Reviejo, M., and Pilling, M. J.: Evaluation of detailed aromatic mechanisms (MCMv3 and MCMv3.1) against environmental chamber data, Atmos. Chem. Phys., 5, 623–639, 2005.

Byron, J., Kreuzwieser, J., Purser, G., van Haren, J., Ladd, S. N., Meredith, L. K., Werner, C., and Williams, J.: Chiral monoterpenes reveal forest emission mechanisms and drought responses, Nature, 609, 307-312, 10.1038/s41586-022-05020-5, 2022.

Chen, J., Liu, T., Gong, D., Li, J., Chen, X., Li, Q., Liao, T., Zhou, Y., Zhang, T., Wang, Y., Wang, H., and Wang, B.: Insight into decreased ozone formation across the Chinese National

- Day Holidays at a regional background site in the Pearl River Delta, Atmos. Environ., 315, 10.1016/j.atmosenv.2023.120142, 2023.
- Chen, T., Xue, L., Zheng, P., Zhang, Y., Liu, Y., Sun, J., Han, G., Li, H., Zhang, X., Li, Y., Li, H., Dong, C., Xu, F., Zhang, Q., and Wang, W.: Volatile organic compounds and ozone air pollution in an oil production region in northern China, Atmos. Chem. Phys., 20, 7069-7086, 10.5194/acp-20-7069-2020, 2020.
- Claflin, M. S., Pagonis, D., Finewax, Z., Handschy, A. V., Day, D. A., Brown, W. L., Jayne, J. T., Worsnop, D. R., Jimenez, J. L., Ziemann, P. J., de Gouw, J., and Lerner, B. M.: An in situ gas chromatograph with automatic detector switching between PTR- and EI-TOF-MS: isomerresolved measurements of indoor air, Atmospheric Measurement Techniques, 14, 133-152, 10.5194/amt-14-133-2021, 2021.
- Coggon, M. M., Stockwell, C. E., Claflin, M. S., Pfannerstill, E. Y., Xu, L., Gilman, J. B., Marcantonio, J., Cao, C., Bates, K., Gkatzelis, G. I., Lamplugh, A., Katz, E. F., Arata, C., Apel, E. C., Hornbrook, R. S., Piel, F., Majluf, F., Blake, D. R., Wisthaler, A., Canagaratna, M., Lerner, B. M., Goldstein, A. H., Mak, J. E., and Warneke, C.: Identifying and correcting interferences to PTR-ToF-MS measurements of isoprene and other urban volatile organic compounds, Atmospheric Measurement Techniques, 17, 801-825, 10.5194/amt-17-801-2024, 2024.
- Feng, X., Guo, J., Wang, Z., Gu, D., Ho, K.-F., Chen, Y., Liao, K., Cheung, V. T. F., Louie, P. K. K., Leung, K. K. M., Yu, J. Z., Fung, J. C. H., and Lau, A. K. H.: Investigation of the multi-year trend of surface ozone and ozone-precursor relationship in Hong Kong, Atmos. Environ., 315, 10.1016/j.atmosenv.2023.120139, 2023.
- Han, J., Liu, Z., Hu, B., Zhu, W., Tang, G., Liu, Q., Ji, D., and Wang, Y.: Observations and explicit modeling of summer and autumn ozone formation in urban Beijing: Identification of key precursor species and sources, Atmos. Environ., 309, 10.1016/j.atmosenv.2023.119932, 2023.
- Holzinger, R., Acton, W. J. F., Bloss, W. J., Breitenlechner, M., Crilley, L. R., Dusanter, S., Gonin, M., Gros, V., Keutsch, F. N., Kiendler-Scharr, A., Kramer, L. J., Krechmer, J. E., Languille, B., Locoge, N., Lopez-Hilfiker, F., Materic, D., Moreno, S., Nemitz, E., Quelever, L. L. J., Esteve, R. S., Sauvage, S., Schallhart, S., Sommariva, R., Tillmann, R., Wedel, S., Worton, D. R., Xu, K. M., and Zaytsev, A.: Validity and limitations of simple reaction kinetics to calculate concentrations of organic compounds from ion counts in PTR-MS, Atmospheric Measurement Techniques, 12, 6193-6208, 10.5194/amt-12-6193-2019, 2019.
- Huang, W., Zhao, Q., Liu, Q., Chen, F., He, Z., Guo, H., and Ling, Z.: Assessment of atmospheric photochemical reactivity in the Yangtze River Delta using a photochemical box model, Atmospheric Research, 245, 10.1016/j.atmosres.2020.105088, 2020.
- Kammer, J., Flaud, P. M., Chazeaubeny, A., Ciuraru, R., Le Menach, K., Geneste, E., Budzinski, H., Bonnefond, J. M., Lamaud, E., Perraudin, E., and Villenave, E.: Biogenic volatile organic compounds (BVOCs) reactivity related to new particle formation (NPF) over the Landes forest, Atmospheric Research, 237, 10.1016/j.atmosres.2020.104869, 2020.
- Karl, T., Striednig, M., Graus, M., Hammerle, A., and Wohlfahrt, G.: Urban flux measurements reveal a large pool of oxygenated volatile organic compound emissions, Proc Natl Acad Sci U S A, 115, 1186-1191, 10.1073/pnas.1714715115, 2018.
- Kim, S., Karl, T., Helmig, D., Daly, R., Rasmussen, R., and Guenther, A.: Measurement of atmospheric sesquiterpenes by proton transfer reaction-mass spectrometry (PTR-MS), Atmos. Meas. Tech., 2, 99–112, 2009.
- Koss, A. R., Sekimoto, K., Gilman, J. B., Selimovic, V., Coggon, M. M., Zarzana, K. J., Yuan, B., Lerner, B. M., Brown, S. S., Jimenez, J. L., Krechmer, J., Roberts, J. M., Warneke, C., Yokelson, R. J., and de Gouw, J.: Non-methane organic gas emissions from biomass burning: identification, quantification, and emission factors from PTR-ToF during the FIREX 2016 laboratory experiment, Atmos. Chem. Phys., 18, 3299-3319, 10.5194/acp-18-3299-2018, 2018.

- Li, Z., Lau, W. K. M., Ramanathan, V., Wu, G., Ding, Y., Manoj, M. G., Liu, J., Qian, Y., Li, J., Zhou, T., Fan, J., Rosenfeld, D., Ming, Y., Wang, Y., Huang, J., Wang, B., Xu, X., Lee, S. S., Cribb, M., Zhang, F., Yang, X., Zhao, C., Takemura, T., Wang, K., Xia, X., Yin, Y., Zhang, H., Guo, J., Zhai, P. M., Sugimoto, N., Babu, S. S., and Brasseur, G. P.: Aerosol and monsoon climate interactions over Asia, Reviews of Geophysics, 54, 866-929, 10.1002/2015rg000500, 2016.
- Liu, X., Lyu, X., Wang, Y., Jiang, F., and Guo, H.: Intercomparison of O3 formation and radical chemistry in the past decade at a suburban site in Hong Kong, Atmos. Chem. Phys., 19, 5127-5145, 10.5194/acp-19-5127-2019, 2019.
- Liu, Y., Qiu, P., Li, C., Li, X., Ma, W., Yin, S., Yu, Q., Li, J., and Liu, X.: Evolution and variations of atmospheric VOCs and O3 photochemistry during a summer O3 event in a county-level city, Southern China, Atmos. Environ., 272, 10.1016/j.atmosenv.2022.118942, 2022.
- Mo, Z., Shao, M., and Lu, S.: Compilation of a source profile database for hydrocarbon and OVOC emissions in China, Atmos. Environ., 143, 209-217, 10.1016/j.atmosenv.2016.08.025, 2016.
- Sarkar, C., Sinha, V., Kumar, V., Rupakheti, M., Panday, A., Mahata, K. S., Rupakheti, D., Kathayat, B., and Lawrence, M. G.: Overview of VOC emissions and chemistry from PTR-TOF-MS measurements during the SusKat-ABC campaign: high acetaldehyde, isoprene and isocyanic acid in wintertime air of the Kathmandu Valley, Atmos. Chem. Phys., 16, 3979-4003, DOI 10.5194/acp-16-3979-2016, 2016.
- Sekimoto, K., Li, S.-M., Yuan, B., Koss, A., Coggon, M., Warneke, C., and de Gouw, J.: Calculation of the sensitivity of proton-transfer-reaction mass spectrometry (PTR-MS) for organic trace gases using molecular properties, International Journal of Mass Spectrometry, 421, 71-94, <a href="https://doi.org/10.1016/j.ijms.2017.04.006">https://doi.org/10.1016/j.ijms.2017.04.006</a>, 2017.
- Shen, H., Liu, Y., Zhao, M., Li, J., Zhang, Y., Yang, J., Jiang, Y., Chen, T., Chen, M., Huang, X., Li, C., Guo, D., Sun, X., Xue, L., and Wang, W.: Significance of carbonyl compounds to photochemical ozone formation in a coastal city (Shantou) in eastern China, Sci Total Environ, 764, 144031, 10.1016/j.scitotenv.2020.144031, 2021.
- Wang, W., Yuan, B., Peng, Y., Su, H., Cheng, Y., Yang, S., Wu, C., Qi, J., Bao, F., Huangfu, Y., Wang, C., Ye, C., Wang, Z., Wang, B., Wang, X., Song, W., Hu, W., Cheng, P., Zhu, M., Zheng, J., and Shao, M.: Direct observations indicate photodegradable oxygenated volatile organic compounds (OVOCs) as larger contributors to radicals and ozone production in the atmosphere, Atmos. Chem. Phys., 22, 4117-4128, 10.5194/acp-22-4117-2022, 2022.
- Whalley, L. K., Stone, D., Dunmore, R., Hamilton, J., Hopkins, J. R., Lee, J. D., Lewis, A. C., Williams, P., Kleffmann, J., Laufs, S., Woodward-Massey, R., and Heard, D. E.: Understanding in situ ozone production in the summertime through radical observations and modelling studies during the Clean air for London project (ClearfLo), Atmos. Chem. Phys., 18, 2547-2571, 10.5194/acp-18-2547-2018, 2018.
- Whalley, L. K., Slater, E. J., Woodward-Massey, R., Ye, C., Lee, J. D., Squires, F., Hopkins, J. R., Dunmore, R. E., Shaw, M., Hamilton, J. F., Lewis, A. C., Mehra, A., Worrall, S. D., Bacak, A., Bannan, T. J., Coe, H., Percival, C. J., Ouyang, B., Jones, R. L., Crilley, L. R., Kramer, L. J., Bloss, W. J., Vu, T., Kotthaus, S., Grimmond, S., Sun, Y., Xu, W., Yue, S., Ren, L., Acton, W. J. F., Hewitt, C. N., Wang, X., Fu, P., and Heard, D. E.: Evaluating the sensitivity of radical chemistry and ozone formation to ambient VOCs and NOx in Beijing, Atmos. Chem. Phys., 21, 2125-2147, 10.5194/acp-21-2125-2021, 2021.
- Wong, Y. K., Liu, K. M., Yeung, C., Leung, K. K. M., and Yu, J. Z.: Measurement report: Characterization and source apportionment of coarse particulate matter in Hong Kong: insights into the constituents of unidentified mass and source origins in a coastal city in southern China, Atmos. Chem. Phys., 22, 5017-5031, 10.5194/acp-22-5017-2022, 2022.
- Wu, C., Wang, C., Wang, S., Wang, W., Yuan, B., Qi, J., Wang, B., Wang, H., Wang, C., Song,

- W., Wang, X., Hu, W., Lou, S., Ye, C., Peng, Y., Wang, Z., Huangfu, Y., Xie, Y., Zhu, M., Zheng, J., Wang, X., Jiang, B., Zhang, Z., and Shao, M.: Measurement report: Important contributions of oxygenated compounds to emissions and chemistry of volatile organic compounds in urban air, Atmos. Chem. Phys., 20, 14769-14785, 10.5194/acp-20-14769-2020, 2020.
- Yang, X., Xue, L., Wang, T., Wang, X., Gao, J., Lee, S., Blake, D. R., Chai, F., and Wang, W.: Observations and Explicit Modeling of Summertime Carbonyl Formation in Beijing: Identification of Key Precursor Species and Their Impact on Atmospheric Oxidation Chemistry, Journal of Geophysical Research: Atmospheres, 123, 1426-1440, 10.1002/2017jd027403, 2018. Yu, D., Tan, Z., Lu, K., Ma, X., Li, X., Chen, S., Zhu, B., Lin, L., Li, Y., Qiu, P., Yang, X., Liu, Y., Wang, H., He, L., Huang, X., and Zhang, Y.: An explicit study of local ozone budget and NOx-VOCs sensitivity in Shenzhen China, Atmos. Environ., 224, 10.1016/j.atmosenv.2020.117304, 2020.
- Yuan, B., Koss, A. R., Warneke, C., Coggon, M., Sekimoto, K., and de Gouw, J. A.: Proton-Transfer-Reaction Mass Spectrometry: Applications in Atmospheric Sciences, Chem Rev, 117, 13187-13229, 10.1021/acs.chemrev.7b00325, 2017.
- Zhang, Y., Wang, Y., Li, C., Li, Y., Yin, S., Claflin, M. S., Lerner, B. M., Worsnop, D., and Wang, L.: Interpretation of mass spectra by a Vocus proton-transfer-reaction mass spectrometer (PTR-MS) at an urban site: insights from gas chromatographic pre-separation, Atmospheric Measurement Techniques, 18, 3547-3568, 10.5194/amt-18-3547-2025, 2025.
- Zhang, Z., Man, H., Duan, F., Lv, Z., Zheng, S., Zhao, J., Huang, F., Luo, Z., He, K., and Liu, H.: Evaluation of the VOC pollution pattern and emission characteristics during the Beijing resurgence of COVID-19 in summer 2020 based on the measurement of PTR-ToF-MS, Environ Res Lett, 17, 10.1088/1748-9326/ac3e99, 2022.
- Zhao, Y., Chen, L., Li, K., Han, L., Zhang, X., Wu, X., Gao, X., Azzi, M., and Cen, K.: Atmospheric ozone chemistry and control strategies in Hangzhou, China: Application of a 0-D box model, Atmospheric Research, 246, 10.1016/j.atmosres.2020.105109, 2020.
- Zhou, X., Li, Z. Q., Zhang, T. J., Wang, F., Wang, F. T., Tao, Y., Zhang, X., Wang, F. L., and Huang, J.: Volatile organic compounds in a typical petrochemical industrialized valley city of northwest China based on high-resolution PTR-MS measurements: Characterization, sources and chemical effects, Sci. Total Environ., 671, 883-896, 10.1016/j.scitotenv.2019.03.283, 2019.

---

## Author Comment (AC2)

**Response to Referee #2**

Thanks very much for your time reviewing our manuscript entitled "The critical role of oxygenated volatile organic compounds (OVOCs) in shaping photochemical O3 chemistry and control strategy in a subtropical coastal environment". We are very grateful to the reviewers for their valuable and helpful suggestions for our manuscript. We have made all the suggested changes and clarifications. We believe that the manuscript has been significantly improved based on those suggestions.

Our point-by-point responses to reviewers' comments are as follows. We repeat the comments raised by the reviewers in *black italic* font and give our replies in the indent and normal font, and with the revised text in blue. The line numbers mentioned in responses correspond to the revised manuscript.

**Evaluation:** "The critical role of oxygenated volatile organic compounds (OVOCs) in shaping photochemical O3 chemistry and control strategy in a subtropical coastal environment" presents a comprehensive and well written box modelling study of the drivers of photochemical O3 formation at a subtropical coastal region in South China, particularly highlighting the importance of including a detailed representation of oxygenated VOCs (OVOCs) in the modelling study. An OVOC sensitivity study is presented, including model results with and without detailed OVOC inclusion, relevant to the broader photochemical O3 production community. I would recommend this manuscript for publication, provided a few further questions are addressed.

**Response:** We appreciate the positive comments from the reviewer. We have revised the manuscript and addressed the issues according to the reviewer's comments.

1: The authors present two case studies: Box modelling with and without the inclusion of OVOCs, and the impact this has on modelling photochemical O3 production. However, in reality, it is more typical for studies to include a subset of OVOCs (though not 63, as is the case in your study), since many of these species can be measured using GC techniques alone. It would be interesting to know how good a job your box modelling does if just a "common" subset of OVOCs is incorporated (e.g. acetone, methanol, acetaldehyde). Does this subset sufficiently describe the ozone production regime? Or is it necessary to include a much more comprehensive suite of OVOCs to fully describe the chemistry? It would be worth taking a look at some literature to see which OVOCs are typically included in modelling studies for this analysis. Some examples: Whalley et al., 2018, Whalley et al., 2021, Nelson et al., 2021. In the absence of re-running the model using a hybrid between no OVOCs and 63 OVOCs, perhaps the authors already have enough information from their current model study to comment on which OVOCs are particularly important to include?

**Response:** We thank the reviewer for the insightful comment regarding the representativeness of OVOCs in photochemical modelling. Based on our review of previous modeling studies and comparison with OVOCs measured by the PTR in this work, we found that typical model inputs of OVOCs often include species such as acetaldehyde, acetone, MEK, and butanal (Whalley et al., 2021; Whalley et al., 2018; Yang et al., 2018; Feng et al., 2023; Shen et al., 2021). To assess whether this subset sufficiently captures the role of OVOCs in O3 and ROx radical production, we conducted an additional sensitivity case in which only these four OVOC species were included in the constrained inputs. We then compared the production rates of O3, OH, HO2, and RO2 with the cases of full OVOCs constraints and without OVOCs constraints. The results show that with only these four OVOCs led to an increase of 12.5%-23.9% in daytime O3 and

ROx radical production rates compared to the case without OVOCs constraints, accounting for around 60% of the underestimation compared to considering all measured OVOCs. This indicates that the significant underestimation of O3 and radical production rates without OVOCs constraints cannot be explained solely by the common subset of OVOC species. In other words, approximately 33.9%-36.5% of photochemical production potential arises from a broader range of OVOCs beyond the commonly OVOCs. This highlights the importance of incorporating a more comprehensive suite of OVOCs in atmospheric chemical modeling for accurately representing O3 formation and radical chemistry in the atmosphere. We have added this new sensitivity case to the revised manuscript to strengthen the discussion on the importance of more comprehensive OVOCs in photochemical modeling.

**The revised text reads:**

**L517-L522**: A sensitivity test considering only four simple OVOCs including acetaldehyde, acetone, MEK, and butanal (Whalley et al., 2021; Whalley et al., 2018; Yang et al., 2018; Feng et al., 2023; Shen et al., 2021) resulted in an increase of 12.5%-23.9% in daytime O3 and ROx production rates than case without any OVOC constrains. This indicates that nearly 33.9%-36.5% reduction cannot be explained by these simple OVOCs alone, further highlighting the importance of comprehensive OVOCs in modeling.

**2:** Line 95 – Is it correct to say that summer is Sept 4 – Oct 12, autumn is Oct 13 – Dec 1, and winter is Dec 2 – Dec 20 in this region? This is quite a short timeframe to cover these seasons. How representative do you feel these windows are of the "summer", "autumn", and "winter" in South China?

**Response:** We appreciate the reviewer's comment. Since Hong Kong is located in a subtropical region where seasonal weather patterns are influenced by the Asian monsoon system. As a result, the direction of upper-level winds is commonly used to characterize seasonal transitions in this region. In this study, the seasonal classification follows the approach adopted in previous studies (Feng et al., 2023), which is based on the occurrence of synoptic-scale events and abrupt changes in key meteorological parameters, including upper-level wind direction, sea-level pressure, and dew point temperature. The temporal variation of upper-level wind direction, sea level pressure and dew point in Hong Kong measured by Hong Kong Observatory Station from July 2021 to March 2022 was shown in Figure S1. A sudden increase in sea-level pressure accompanied by a notable decrease in dew point on 12 Oct 2021 indicated intrusion of relatively cold air masses, making the transition from summer to autumn. During the summer period, upper-level winds were predominantly easterly and/or southeasterly, reflecting typical monsoonal circulation patterns (Wong et al., 2022). Similarly, the transition from autumn to winter was characterized by abrupt changes in both dew point and upper-level wind direction, with the characteristics of cold high-pressure climate. Moreover, in winter, upper-level winds were primarily westerly and/or northwestly, consistent with the influence of the East Asian winter monsoon (Wong et al., 2022; Li et al., 2016). Therefore, the seasonal division and selected period in this study is considered representative of the characteristic features of summer, autumn, and early winter.

**The revised text reads:**

**L96-L99**: Seasonal classification in this study was based on the occurrence of synoptic events and abrupt changes in key meteorological parameters, including upper-level wind direction, sea-level pressure, and dew point temperature (Figure S1), as detailed in our previous studies (Feng et al., 2023).

Figure S1. Temporal variation of upper-level wind direction, sea level pressure and dew point in Hong Kong measured by Hong Kong Observatory Station from July 2021 to March 2022. The seasonal transition from summer to early winter was characterized by a rapid shift from high dew point and low sea-level pressure to cold and high-pressure systems, accompanied by a change in upper-level wind direction from easterly/southeasterly to westerly/northwesterly (Li et al., 2016; Wong et al., 2022).

**3:** Line 135 – We "attempted" to assign signals based on likely contributors, and that your quantification of OVOCs is semi-quantitative for uncalibrated species, is an honest and fair account of what you have done here. Whilst I completely understand that this must be the case due to instrument limitations, some readers may feel a little mis-sold up to this point, as the abstract implies that we are to expect 63 explicitly measured and quantified OVOCs. Perhaps you could be more upfront in the abstract on what you have done to describe the OVOCs, as this is a major component of your study. Even if you just said "X quantified and Y semi-quantified OVOC species", rather than 63 quantified (abstract – line 22).

**Response:** Thanks for the valuable suggestion. We have revised the 63 quantified OVOC species to 8 quantified and 55 semi-quantified OVOC species in the abstract and the manuscript.

**L175-L178**: Given these limitations, our quantification of the 63 OVOCs measured by PTR should be considered as semi-quantitative for 55 uncalibrated species, and as high-confidence for the 8 calibrated species.

**4:** Line 160 – I was struggling to follow exactly what you've done here. "VOC species from daytime canister samples were linearly interpolated to hourly resolution for the model input". Is this just the VOCs measured using the GC-MS/FID/ECD? I assume the PTR measurements are online? Please be clear on the time resolution of the measurement of both things. How often were canister samples taken? Is interpolating the data to hourly resolution appropriate for this measurement resolution?

**Response:** Thanks for the valuable comment. We apologize for the confusion and have revised the manuscript to clarify the data sources and their temporal resolutions. Please find our responses below:

The VOC species used for linear interpolation were measured using offline canister sampling and analyzed by GC-MS/FID/ECD. These measurements primarily included 38 species including alkanes, alkenes, alkynes, and alkyl nitrates, which were used as input for model

simulations. These canister samples were collected every 3 hours from 09:00 to 18:00 LT during the selected polluted days. Given the limited temporal resolution of the offline sampling method, a linear interpolation approach was applied to estimate hourly concentrations during the daytime, following methodologies adopted in previous studies (Yang et al., 2018). While the nighttime concentrations were estimated using linear regression relationships with continuously measured tracers obtained from PTR-ToF-MS measurements. In contrast, continuous online measurements of ambient VOCs were conducted in this study using PTR-ToF-MS, with a time resolution of 10 seconds. However, we opted not to interpolate the offline data to a resolution finer than one hour, as doing so would likely introduce greater uncertainty and potentially compromise the accuracy and reliability of the model simulations.

**The revised text reads:**

L103-L105: A PTR-ToF-MS (Ionicon Analytik GmbH, Innsbruck, Austria) with H3O+ as the primary reaction ion was used to measure the gaseous VOC and OVOC species with high time resolution of 10 seconds during the whole field campaign.

**L179-L180**: In addition, canister samples of VOCs were collected every 3 hours from 09:00 to 18:00 LT in three seasons.

**L202-L210**: For the offline canister VOC samples measured by GC-MS/FID/ECD, daytime data from 9:00 to 18:00 were linearly interpolated to an hourly resolution for the model input (Yang et al., 2018), while nighttime concentrations of unmeasured  $C_2$ - $C_{10}$  hydrocarbons (excluding isoprene and monoterpenes) and alkyl nitrates were estimated using linear regression relationships with continuously measured hydrocarbons (e.g.,  $C_3H_6$ ,  $C_5H_{10}$ ,  $C_6H_{10}$ ) and nitrophenols obtained from PTR-ToF-MS measurements. The PTR measured species used in the linear regression calculation were selected based on their strong correlations with corresponding compounds in the canister data to ensure more reliable estimates.

**5:** 225 – A lot of effort has been put into speciating the OVOCs, but what about the biogenic species? The authors state there were measurements made of 2 biogenic species – isoprene, and monoterpenes. However, there are only 3 monoterpenes in the MCM (the pinenes and limonene), and they later describe their monoterpene measurement as the pinenes only. As these are highly reactive species, it is important to discuss how your assignment of the total monoterpene measurement to (what I assume is 50:50?) a- and b- pinene. There are also many monoterpene species not included in the MCM, some with faster reaction rates than these species. I understand the need for your assumption to be made, but please acknowledge the potential implications this has in the text.

Response: Thanks for the valuable suggestion. As noted by the reviewer, biogenic species are highly reactive and play a critical role in atmospheric photochemical processes. In this study, biogenic species measured by PTR-ToF-MS mainly included C5H8 and C10H16. C5H8H+ signal mainly corresponds to isoprene and could also be interfered with fragments from higher-carbon aldehydes and cycloalkanes, while C10H16H+ signal primarily corresponds to various kinds of monoterpenes, such as pinenes, limonene, camphene, and so on (Coggon et al., 2024; Claflin et al., 2021; Yuan et al., 2017; Zhang et al., 2025). Moreover, the interferences of fragmentation and the distributions of isomers may vary with altitudes and environmental conditions influenced by both biogenic and anthropogenic emission sources, which makes the isomer distribution more complex and introduces large uncertainties (Coggon et al., 2024). However, due to the limitations of PTR-ToF-MS, accurate quantification of isomers cannot be achieved at present. Therefore, assumptions regarding the distribution of isomers and potential fragment interferences were made based on previous studies using the gas chromatography preseparation. For the attribution of C5H8 signals, isoprene was allocated the fraction of 63% reported in previous studies employing PTR-MS measurements coupled with GC, which effectively minimizes interference from fragments of higher molecular compounds (Koss et al., 2018). While for the distribution of  $C_{10}H_{16}H^+$  signals, based on previous studies,  $\alpha$ - and  $\beta$ pinene generally accounted for the most to  $C_{10}H_{16}H^+$  signals in ambient air, with  $\alpha$ -pinene as the most contributor (Kim et al., 2009; Byron et al., 2022; Kammer et al., 2020). Therefore, in this study, as the reviewer pointed out, we allocated equally between  $\alpha$ -pinene and  $\beta$ -pinene at a 50%:50% ratio as the assumption for model simulation.  $\alpha$ - and  $\beta$ -pinenes are highly reactive species which can rapidly react with atmospheric oxidants such as OH radicals, however, their reactivity remains lower than that of certain other C10H16 species, such as limonene (Bouvier-Brown et al., 2009). Therefore, simplifying the assumption of representing all C10H16 solely by α- and β-pinenes may introduce some uncertainties in the simulation of atmospheric photochemical processes, particularly affecting the simulation of radical budgets and O3 production. Moreover, the absence of other highly reactive monoterpenes and their associated oxidation mechanisms in the MCM may further compromise the accuracy of photochemical simulations. Incorporating a more comprehensive representation of monoterpene chemistry is thus essential for improving the reliability of future atmospheric modeling efforts.

**The revised text reads:**

L169-L174: C5H8 may be affected by fragment interferences from higher-carbon aldehydes and cycloalkanes (Coggon et al., 2024; Claflin et al., 2021; Yuan et al., 2017; Zhang et al., 2025), therefore, the attribution of C5H8 to isoprene follows the proportion of 63% reported in previous studies employing PTR-MS coupled with GC pre-separation, which effectively minimizes interference from fragments of higher molecular compounds (Koss et al., 2018).

**L152-L155**: For example, for  $C_{10}H_{16}$ , given that  $\alpha$ -pinene and  $\beta$ -pinene are typically the predominant contributors (Kim et al., 2009; Byron et al., 2022; Kammer et al., 2020), an equal 50:50 allocation between the two species was adopted as a modeling assumption for the apportionment.

**L280-L285**: BVOCs refer specifically to C5H8 and C10H16, which primarily correspond to isoprene and various monoterpenes (e.g., pinenes, limonene, camphene, etc.), respectively. It is important to note that C5H8 may be affected by fragment interferences from higher-carbon aldehydes and cycloalkanes (Coggon et al., 2024; Claflin et al., 2021; Yuan et al., 2017; Zhang et al., 2025), which may potentially lead to an overestimation of the contribution of BVOCs, particularly isoprene.

**L369-L372**: Nevertheless, due to the inherent limitations of PTR-ToF-MS, accurate quantification of isomers with distinct chemical reactivities remains challenging, introducing some uncertainties in atmospheric photochemical modeling.

**6:** 300 – Just to reiterate my early point, the authors say your measurements are high-resolution here, but the resolution needs to be stated more explicitly earlier in the text.

**Response:** Thanks for the reviewer's comment. We have added a description of the 10-second time resolution used in the PTR-ToF-MS measurement to the Method Section (**Lines 104-105**).

7: 344 - More discussion of the implications of BVOCs in your model here – again, worth pointing out that this result is based on the monoterpenes being split between the pinenes only.

**Response:** Thanks for the valuable suggestion. We have added more discussion of the impacts

of BVOCs on model simulations.

The revised text reads:

**L350-L355**: It should be noted that current photochemical models typically represent monoterpenes using only  $\alpha$ -and  $\beta$ -pinenes, neglecting some highly reactive species such as limonene. Moreover, gaps in the MCM, such as the absence of certain highly reactive monoterpenes and associated oxidation pathways, may further introduce uncertainties in assessing the role of BVOCs in atmospheric photochemistry.

**8:** 397 – The authors say that including the OVOCs means that observed O3 was successfully reproduced. Are the observed and model concentrations identical? I would expect some impact from transportation to play a role here, since not all the observed O3 can be expected to be photochemically produced in situ. Please discuss.

Response: Thanks for the reviewer's comment. We apologize for the unclear clarity in our previous explanation. Our main point is that incorporating a broader range of OVOCs into the model will lead to a relatively improved representation of O3 formation compared to simulations including only a limited set of OVOCs or without OVOC species. However, this improvement does not imply a complete agreement between modeled and observed O3 concentrations. As the reviewer pointed out, the model captures only in situ photochemical processes, whereas ambient O3 levels are also influenced by other factors, such as regional transport. Therefore, exact agreement between modeled and observed O3 concentrations is not expected. Indeed, as shown in the time series of comparison (Figure S13), there are many days on which the simulated O3 levels are lower than the observations, even with the inclusion of broader OVOCs in the modeling. This discrepancy is particularly evident during the autumn and early winter, when the influence of Asian monsoon is strong, and during the nighttime when photochemical reactions are minimal.

**The revised text reads:**

**L470-L472**: Incorporating a broader range of OVOCs improved the simulation of O3, particularly in autumn and early winter, where daytime concentrations were underestimated by 26.5% and 35.7%, respectively, without OVOCs constraints.

L474-L478: It should be noted that the model considers only in situ photochemical processes and does not include influences such as regional transport. As a result, discrepancies between observed and simulated O3 remain, especially in autumn and early winter, when periods typically influenced by the Asian monsoon, and during nighttime when photochemical activity is minimal.

**9:** Line 444 – "our results highlight that many other OVOCs,...,remain overlooked", and line 451 "key OVOC species such as methanol, acetaldehyde, and acetone,..., were underestimated". These key OVOCs are more typically incorporated into box modelling studies in the literature, as they can be measured using GC techniques (see my main suggestion earlier in this review).

**Response:** We thank the reviewer for the valuable comment and apologize for the lack of clarity in our original description. What we intended to emphasize is that, under the scenario without OVOC constraints, the primary reason for the underestimation of O3 and ROx production is the substantial underestimation of OVOC concentrations in the model simulation. The reference to "methanol, acetaldehyde, and acetone being underestimated by 73%-99%"

was intended merely as an example to illustrate this issue. Our intention was not to imply that these compounds are absent from other models reported in the literature.

The revised text reads:

**L525-L528**: These discrepancies were largely due to the underestimation of multiple OVOC species, for example, methanol, acetaldehyde, and acetone were underestimated by 73%-99% in early winter simulations without OVOC constraints (Table S5).

**10:** Line 494 – When you vary the VOCs and NOx for the isopleth analysis, do you also vary OVOCs? The difficulty here is that some of your OVOCs will be formed photochemically, and some from primary sources, meaning the two will not necessarily ever decrease or increase uniformly. Do you also reduce biogenic species, or just anthropogenic? How do you navigate this issue to make the isopleth findings relevant from a policy perspective?

**Response:**

We thank the reviewer for raising this important point regarding the treatment of OVOCs in O3 isopleth analysis and its implications for policy relevance. We acknowledge that the diverse sources of OVOCs (including both primary emissions and secondary formation) present a significant challenge in accurately representing their variability in sensitivity analysis. Due to the current lack of a more robust framework to account for the dynamic behaviors of OVOCs under different emission scenarios, we adopted a simplified approach in this study. Specifically, with OVOCs constraints, changes in OVOC concentrations were scaled proportionally to the variations in their precursor VOCs. This assumption reflects a linear relationship between precursor VOC levels and secondary OVOC formation, providing an approximation of their response to emission changes. While we recognize that this approach does not fully capture the nonlinear chemistry involved in secondary OVOC production, it offers a pragmatic representation of their influence on ozone formation under varying emission conditions. We consider this a reasonable interim approach, pending further advancement in the mechanistic understanding and modeling of OVOC chemistry. And we added relevant clarification of OVOCs in this section.

Regarding biogenic VOCs, we applied uniform scaling to both anthropogenic and biogenic VOCs in the isopleth analysis to explore the full spectrum of chemical regimes and assess O3 formation sensitivity under varying precursor conditions. However, emissions of biogenic VOCs are predominantly driven by natural processes and are not readily amenable to direct control. Therefore, scenarios involving reductions in biogenic VOCs are not intended to reflect realistic policy interventions, but serve as sensitivity tests to better understand the chemical regime transitions and the role of biogenic VOCs in O3 production. For policy-relevant interpretation, we place emphasis on anthropogenic VOC reductions, which are more feasible and actionable from an emissions control perspective.

**The revised text reads:**

**L571-L575:** It should be noted that due to the absence of a robust mechanism to represent the nonlinear formation and diverse sources of OVOCs, we employed a simplified scaling approach based on precursor VOCs in O3 isopleth analysis. Despite inherent uncertainties, this provides a practical approximation for assessing OVOC impacts on O3 formation under varying emission scenarios.

**References:**

Bouvier-Brown, N. C., Goldstein, A. H., Gilman, J. B., Kuster, W. C., and Gouw, J. A. d.: Insitu ambient quantification of monoterpenes, sesquiterpenes, and related oxygenated compounds during BEARPEX 2007: implications for gas- and particle-phase chemistry, Atmos. Chem. Phys., 9, 5505–5518, 2009.

Byron, J., Kreuzwieser, J., Purser, G., van Haren, J., Ladd, S. N., Meredith, L. K., Werner, C., and Williams, J.: Chiral monoterpenes reveal forest emission mechanisms and drought responses, Nature, 609, 307-312, 10.1038/s41586-022-05020-5, 2022.

Claflin, M. S., Pagonis, D., Finewax, Z., Handschy, A. V., Day, D. A., Brown, W. L., Jayne, J. T., Worsnop, D. R., Jimenez, J. L., Ziemann, P. J., de Gouw, J., and Lerner, B. M.: An in situ gas chromatograph with automatic detector switching between PTR- and EI-TOF-MS: isomerresolved measurements of indoor air, Atmospheric Measurement Techniques, 14, 133-152, 10.5194/amt-14-133-2021, 2021.

Coggon, M. M., Stockwell, C. E., Claflin, M. S., Pfannerstill, E. Y., Xu, L., Gilman, J. B., Marcantonio, J., Cao, C., Bates, K., Gkatzelis, G. I., Lamplugh, A., Katz, E. F., Arata, C., Apel, E. C., Hornbrook, R. S., Piel, F., Majluf, F., Blake, D. R., Wisthaler, A., Canagaratna, M., Lerner, B. M., Goldstein, A. H., Mak, J. E., and Warneke, C.: Identifying and correcting interferences to PTR-ToF-MS measurements of isoprene and other urban volatile organic compounds, Atmospheric Measurement Techniques, 17, 801-825, 10.5194/amt-17-801-2024, 2024.

Feng, X., Guo, J., Wang, Z., Gu, D., Ho, K.-F., Chen, Y., Liao, K., Cheung, V. T. F., Louie, P. K. K., Leung, K. K. M., Yu, J. Z., Fung, J. C. H., and Lau, A. K. H.: Investigation of the multi-year trend of surface ozone and ozone-precursor relationship in Hong Kong, Atmos. Environ., 315, 10.1016/j.atmosenv.2023.120139, 2023.

Kammer, J., Flaud, P. M., Chazeaubeny, A., Ciuraru, R., Le Menach, K., Geneste, E., Budzinski, H., Bonnefond, J. M., Lamaud, E., Perraudin, E., and Villenave, E.: Biogenic volatile organic compounds (BVOCs) reactivity related to new particle formation (NPF) over the Landes forest, Atmospheric Research, 237, 10.1016/j.atmosres.2020.104869, 2020.

Kim, S., Karl, T., Helmig, D., Daly, R., Rasmussen, R., and Guenther, A.: Measurement of atmospheric sesquiterpenes by proton transfer reaction-mass spectrometry (PTR-MS), Atmos. Meas. Tech., 2, 99–112, 2009.

Koss, A. R., Sekimoto, K., Gilman, J. B., Selimovic, V., Coggon, M. M., Zarzana, K. J., Yuan, B., Lerner, B. M., Brown, S. S., Jimenez, J. L., Krechmer, J., Roberts, J. M., Warneke, C., Yokelson, R. J., and de Gouw, J.: Non-methane organic gas emissions from biomass burning: identification, quantification, and emission factors from PTR-ToF during the FIREX 2016 laboratory experiment, Atmos. Chem. Phys., 18, 3299-3319, 10.5194/acp-18-3299-2018, 2018. Li, Z., Lau, W. K. M., Ramanathan, V., Wu, G., Ding, Y., Manoj, M. G., Liu, J., Qian, Y., Li, J., Zhou, T., Fan, J., Rosenfeld, D., Ming, Y., Wang, Y., Huang, J., Wang, B., Xu, X., Lee, S. S., Cribb, M., Zhang, F., Yang, X., Zhao, C., Takemura, T., Wang, K., Xia, X., Yin, Y., Zhang, H., Guo, J., Zhai, P. M., Sugimoto, N., Babu, S. S., and Brasseur, G. P.: Aerosol and monsoon climate interactions over Asia, Reviews of Geophysics, 54, 866-929, 10.1002/2015rg000500, 2016.

Shen, H., Liu, Y., Zhao, M., Li, J., Zhang, Y., Yang, J., Jiang, Y., Chen, T., Chen, M., Huang, X., Li, C., Guo, D., Sun, X., Xue, L., and Wang, W.: Significance of carbonyl compounds to photochemical ozone formation in a coastal city (Shantou) in eastern China, Sci Total Environ, 764, 144031, 10.1016/j.scitotenv.2020.144031, 2021.

Whalley, L. K., Stone, D., Dunmore, R., Hamilton, J., Hopkins, J. R., Lee, J. D., Lewis, A. C., Williams, P., Kleffmann, J., Laufs, S., Woodward-Massey, R., and Heard, D. E.: Understanding in situ ozone production in the summertime through radical observations and modelling studies during the Clean air for London project (ClearfLo), Atmos. Chem. Phys., 18, 2547-2571, 10.5194/acp-18-2547-2018, 2018.

Whalley, L. K., Slater, E. J., Woodward-Massey, R., Ye, C., Lee, J. D., Squires, F., Hopkins, J. R., Dunmore, R. E., Shaw, M., Hamilton, J. F., Lewis, A. C., Mehra, A., Worrall, S. D., Bacak, A., Bannan, T. J., Coe, H., Percival, C. J., Ouyang, B., Jones, R. L., Crilley, L. R., Kramer, L. J., Bloss, W. J., Vu, T., Kotthaus, S., Grimmond, S., Sun, Y., Xu, W., Yue, S., Ren, L., Acton, W. J. F., Hewitt, C. N., Wang, X., Fu, P., and Heard, D. E.: Evaluating the sensitivity of radical chemistry and ozone formation to ambient VOCs and NOx in Beijing, Atmos. Chem. Phys., 21, 2125-2147, 10.5194/acp-21-2125-2021, 2021.

Wong, Y. K., Liu, K. M., Yeung, C., Leung, K. K. M., and Yu, J. Z.: Measurement report: Characterization and source apportionment of coarse particulate matter in Hong Kong: insights into the constituents of unidentified mass and source origins in a coastal city in southern China, Atmos. Chem. Phys., 22, 5017-5031, 10.5194/acp-22-5017-2022, 2022.

Yang, X., Xue, L., Wang, T., Wang, X., Gao, J., Lee, S., Blake, D. R., Chai, F., and Wang, W.: Observations and Explicit Modeling of Summertime Carbonyl Formation in Beijing: Identification of Key Precursor Species and Their Impact on Atmospheric Oxidation Chemistry, Journal of Geophysical Research: Atmospheres, 123, 1426-1440, 10.1002/2017jd027403, 2018. Yuan, B., Koss, A. R., Warneke, C., Coggon, M., Sekimoto, K., and de Gouw, J. A.: Proton-Transfer-Reaction Mass Spectrometry: Applications in Atmospheric Sciences, Chem Rev, 117, 13187-13229, 10.1021/acs.chemrev.7b00325, 2017.

Zhang, Y., Wang, Y., Li, C., Li, Y., Yin, S., Claflin, M. S., Lerner, B. M., Worsnop, D., and Wang, L.: Interpretation of mass spectra by a Vocus proton-transfer-reaction mass spectrometer (PTR-MS) at an urban site: insights from gas chromatographic pre-separation, Atmospheric Measurement Techniques, 18, 3547-3568, 10.5194/amt-18-3547-2025, 2025.